# Targeting a cell surface vitamin D receptor on tumor-associated macrophages in triple-negative breast cancer

Fernanda I Staquicini[1,2‡], Amin Hajitou[3], Wouter HP Driessen[4§], Bettina Proneth[5], Marina Cardó-Vila[6,7], Daniela I Staquicini[1,2], Christopher Markosian[1,2], Maria Hoh[8#], Mauro Cortez[9], Anupama Hooda-Nehra[1,10], Mohammed Jaloudi[1,10], Israel T Silva[11], Jaqueline Buttura[11], Diana N Nunes[12], Emmanuel Dias-Neto[11,12], Bedrich Eckhardt[13], Javier Ruiz-Ramírez[14], Prashant Dogra[14], Zhihui Wang[14], Vittorio Cristini[14], Martin Trepel[15,16], Robin Anderson[13], Richard L Sidman[17], Juri G Gelovani[18,19,20¶], Massimo Cristofanilli[21], Gabriel N Hortobagyi[22], Zaver M Bhujwalla[8], Stephen K Burley[23,24,25], Wadih Arap[1,10†*], Renata Pasqualini[1,2†*]

*For correspondence:
wa116@newark.rutgers.edu (WA);
rp946@rutgers.edu (RP)

†These authors contributed equally to this work

Present address: ‡MBrace Therapeutics, TO Daniel Research Incubator and Collaboration Center, Summit, United States; §Roche Pharma Research and Early Development, Roche Innovation Center Basel, Basel, Switzerland; #Department of Pharmacology, School of Medicine, University of Colorado, Aurora, United States; ¶Office of the Provost, United Arab Emirates University, Al Ain, United Arab Emirates

[1]Rutgers Cancer Institute of New Jersey, Newark, United States; [2]Division of Cancer Biology, Department of Radiation Oncology, Rutgers New Jersey Medical School, Newark, United States; [3]Phage Therapy Group, Department of Brain Sciences, Imperial College London, London, United Kingdom; [4]The University of Texas M.D. Anderson Cancer Center, Houston, United States; [5]Institute of Metabolism and Cell Death, Helmholtz Zentrum Muenchen, Neuherberg, Germany; [6]Department of Cellular and Molecular Medicine, The University of Arizona Cancer Center, University of Arizona, Tucson, United States; [7]Department of Otolaryngology-Head and Neck Surgery, The University of Arizona Cancer Center, University of Arizona, Tucson, United States; [8]Division of Cancer Imaging Research, The Russell H Morgan Department of Radiology and Radiological Science, The Johns Hopkins University School of Medicine, Baltimore, United States; [9]Department of Parasitology, Institute of Biomedical Sciences, University of São Paulo, São Paulo, Brazil; [10]Division of Hematology/Oncology, Department of Medicine, Rutgers New Jersey Medical School, Newark, United States; [11]Laboratory of Computational Biology, A.C. Camargo Cancer Center, São Paulo, Brazil; [12]Laboratory of Medical Genomics, A.C. Camargo Cancer Center, São Paulo, Brazil; [13]Translational Breast Cancer Program, Olivia Newton-John Cancer Research Institute, Melbourne, Australia; [14]Mathematics in Medicine Program, The Houston Methodist Research Institute, Houston, United States; [15]Department of Oncology and Hematology, University Medical Center Hamburg-Eppendorf, Hamburg, Germany; [16]Department of Oncology and Hematology, University Medical Center Augsburg, Augsburg, Germany; [17]Department of Neurology, Harvard Medical School, Boston, United States; [18]Department of Biomedical Engineering, College of Engineering, Wayne State University, Detroit, United States; [19]Department of Oncology, School of Medicine, Wayne State University, Detroit, United States; [20]Department of Neurosurgery, School of Medicine, Wayne State University, Detroit, United States; [21]Robert H Lurie Comprehensive Cancer Center, Feinberg School of Medicine, Northwestern University Chicago, Chicago, United States; [22]Department of Breast Medical Oncology, The University of Texas M.D. Anderson Cancer Center, Houston, United States; [23]Rutgers Cancer Institute of New Jersey,

New Brunswick, United States; [24]Research Collaboratory for Structural Bioinformatics Protein Data Bank, San Diego Supercomputer Center, University of California-San Diego, La Jolla, United States; [25]Research Collaboratory for Structural Bioinformatics Protein Data Bank, Institute for Quantitative Biomedicine, Rutgers, The State University of New Jersey, Piscataway, United States

**Abstract** Triple-negative breast cancer (TNBC) is an aggressive tumor with limited treatment options and poor prognosis. We applied the in vivo phage display technology to isolate peptides homing to the immunosuppressive cellular microenvironment of TNBC as a strategy for non-malignant target discovery. We identified a cyclic peptide (CSSTRESAC) that specifically binds to a vitamin D receptor, protein disulfide-isomerase A3 (PDIA3) expressed on the cell surface of tumor-associated macrophages (TAM), and targets breast cancer in syngeneic TNBC, non-TNBC xenograft, and transgenic mouse models. Systemic administration of CSSTRESAC to TNBC-bearing mice shifted the cytokine profile toward an antitumor immune response and delayed tumor growth. Moreover, CSSTRESAC enabled ligand-directed theranostic delivery to tumors and a mathematical model confirmed our experimental findings. Finally, in silico analysis showed PDIA3-expressing TAM in TNBC patients. This work uncovers a functional interplay between a cell surface vitamin D receptor in TAM and antitumor immune response that could be therapeutically exploited.

## Introduction

Breast cancer is the second most common cancer type worldwide, and triple-negative breast cancer (TNBC) comprises up to ~10–20% of all cases. These heterogeneous tumors are clinically aggressive, usually with larger sizes at initial presentation, of high pathological grade, and likely to have lymph node involvement and early recurrence in visceral sites (*Dietze et al., 2015*; *Newman and Kaljee, 2017*; *Schettini et al., 2016*). TNBC is treated with multimodality therapy including neoadjuvant chemotherapy, surgery and adjuvant radiotherapy, with selected patients receiving additional adjuvant systemic therapy. Despite optimal management, many patients have distant metastases and poor disease outcomes (*Biswas et al., 2016*; *Coughlin, 2019*; *Dent et al., 2007*; *Perou et al., 2000*). Combination chemotherapy has long been the standard therapeutic option but checkpoint inhibitors and poly ADP-ribose polymerase (PARP) inhibitors have recently been approved in certain settings (*Garrido-Castro et al., 2019*; *Khan et al., 2019*; *Lyons and Traina, 2019*; *Marra et al., 2019*).

Immunomodulators are among the best available investigational drugs for this tumor subtype, based on the premise that manipulation of the local and/or distant immune responses may ultimately represent a viable treatment approach (*Marra et al., 2019*). A biological hallmark of TNBC is an immunosuppressive tumor microenvironment that fosters tumor growth and metastatic spread through the suppression of tumor-infiltrating lymphocytes and secretion of immunoinhibitory cytokines, mainly by tumor-associated macrophages (TAM) (*DeNardo and Ruffell, 2019*; *Lim et al., 2018*; *Wagner et al., 2019*). TAM are classically divided into two major populations, M1 and M2, representing the extremes of a broad activation state spectrum; the M1 population is associated with antitumor activity while the M2 population with tumor progression (*DeNardo and Ruffell, 2019*; *Lim et al., 2018*; *Wagner et al., 2019*; *Biswas and Mantovani, 2010*; *Tan et al., 2019*). Such biological behavior in breast cancer has made them potentially attractive targets for therapeutic intervention. In fact, TAM-targeting drugs are currently in clinical trials but have not yet been approved for clinical practice.

## Results

### Combinatorial phage display screening in vivo reveals tumor microenvironment-binding peptides in a mouse model of TNBC

We used a phage display-based approach to identify homing peptides that target TAM in TNBC. The EF43.*fgf4* syngeneic mouse mammary gland tumor (*Adams et al., 1987*; *Hajitou et al., 1998*) is highly infiltrated by TAM and also serves as an immunocompetent TNBC model since EF43.*fgf4* cells

do not express the estrogen receptor, progesterone receptor, or *Erbb2*/Neu (*Figure 1—figure supplement 1A*). A random phage peptide library was first administered intravenously (iv) in immunocompetent female BALB/c mice with established EF43.*fgf4*-derived mammary fat pad tumors. Phage particles were recovered from tumors after 24 hr, re-amplified, and subjected to two additional rounds of in vivo selection. After the third round, the pool of tumor-homing phage showed an ~300 fold enrichment relative to normal tissues (*Figure 1A*). Bioinformatic analysis of peptides targeting the whole tumor revealed four sequences above an experimental threshold (set at 1%): CSSTRESAC, CRYSAARSC, CRGFVVGRC, and CQRALMIAC (*Figure 1—figure supplement 1B*). Notably, the dominant peptide CSSTRESAC was more strongly enriched (16-fold) than each of the other three peptides (*Figure 1B*). The four selected peptides were next individually evaluated based on absence of binding to EF43.*fgf4* cells in vitro (*Figure 1B*). With a standard cell binding assay (*Giordano et al., 2001*), we found that the peptides CRGFVVGRC, CQRALMIAC, and CRYSAARSC bound to EF43.*fgf4* cells, whereas the peptide CSSTRESAC did not (*Figure 1B*), indicating that CSSTRESAC might indeed recognize non-malignant stromal cells within the tumor microenvironment. The peptides CRGFVVGRC, CQRALMIAC, and CRYSAARSC were not studied further.

To identify the non-malignant cellular component(s) targeted by CSSTRESAC-phage, we tested binding to subcellular populations freshly isolated from engrafted tumors. mCherry-expressing EF43.*fgf4* cells were FACS-sorted from whole tumors. The remaining cells were subsequently FACS-sorted based on expression of CD45 (Leukocyte Common Antigen, LCA) and F4/80, respectively. Similar to human breast cancers known to be highly infiltrated by macrophages (*Biswas et al., 2016*; *Dent et al., 2007*; *Perou et al., 2000*; *Garrido-Castro et al., 2019*; *Khan et al., 2019*; *Marra et al., 2019*; *DeNardo and Ruffell, 2019*; *Lim et al., 2018*; *Wagner et al., 2019*; *Biswas and Mantovani, 2010*; *Tan et al., 2019*; *Adams et al., 1987*; *Hajitou et al., 1998*; *Giordano et al., 2001*), the macrophage population (CD11b$^+$F4/80$^+$) constituted a large portion of the non-malignant cellular component of EF43.*fgf4*-derived mammary tumors, followed by a lesser population of B lymphocytes (CD45R$^+$). T-lymphocytes (CD8$^+$ or CD4$^+$) were not detected (*Figure 1—figure supplement 2*). Binding assays to each of these cell subpopulations showed that CSSTRESAC-phage particles bound specifically to CD11b$^+$F4/80$^+$ macrophage; binding to tumor-isolated EF43.*fgf4* cells and CD45R$^+$ cells were at background levels (*Figure 1C*). Based on these results, we concluded that CSSTRESAC-phage particles target a TAM cell surface receptor.

Although we showed that the CSSTRESAC-phage targeted TAM in a syngeneic TNBC model, we considered that it might be able to target the tumor microenvironment in other experimental models of non-TNBC breast cancer also known to be infiltrated by TAM. First, we tested CSSTRESAC-phage homing in the mouse mammary tumor virus-polyoma middle T-antigen (MMTV-PyMT) transgenic model of breast cancer (*Guy et al., 1992*; *Maglione et al., 2001*). Binding of the CSSTRESAC-phage to MMTV-PyMT tumors was higher compared to a control organ (~3-fold) or to a negative control phage (~2.5-fold) (*Figure 1—figure supplement 3A*). To determine whether the CSSTRESAC-phage may also target human tumors, we next used MDA-MB-231-bearing mice, a standard non-TNBC breast cancer xenograft model. We tested whether liposomes decorated with either CSSTRESAC or control peptide could target these tumors by using Magnetic Resonance Imaging (MRI) and fluorescence, and found that CSSTRESAC targets human breast cancers in vivo independently of the phage context. (*Figure 1—figure supplement 3B–F*). Together, these experiments demonstrate that CSSTRESAC targets a range of different breast tumors (in xenograft, genetic, and syngeneic mouse models) independently of their ligand display context, tumor cell species, or host immunocompetency status. These results also further indicate that the CSSTRESAC peptide may be of value in different types of non-TNBC, and perhaps also other solid tumors containing TAM. Liposome uptake was low in all organs except the liver, a well-known biological phenomenon due to the relatively large size and cationic charge of liposomes (both, targeted or control). None of the liposome preparations caused liver toxicity as confirmed by levels of alanine aminotransferase (ALT) and aspartate aminotransferase (AST) measured in serum of treated mice (*Figure 1—figure supplement 4*).

Next, we used peptide affinity chromatography (*Staquicini et al., 2009*) to identify the cell surface receptor(s) in TAM targeted by the CSSTRESAC peptide. Interacting proteins were eluted through an excessive amount of soluble CSSTRESAC peptide and subsequently control acidic glycine buffer. Binding assays were used to identify eluted fractions containing the highest concentrations of receptor(s) (*Figure 1D*). Proteins present in fraction (F)#5 (positive experimental fraction)

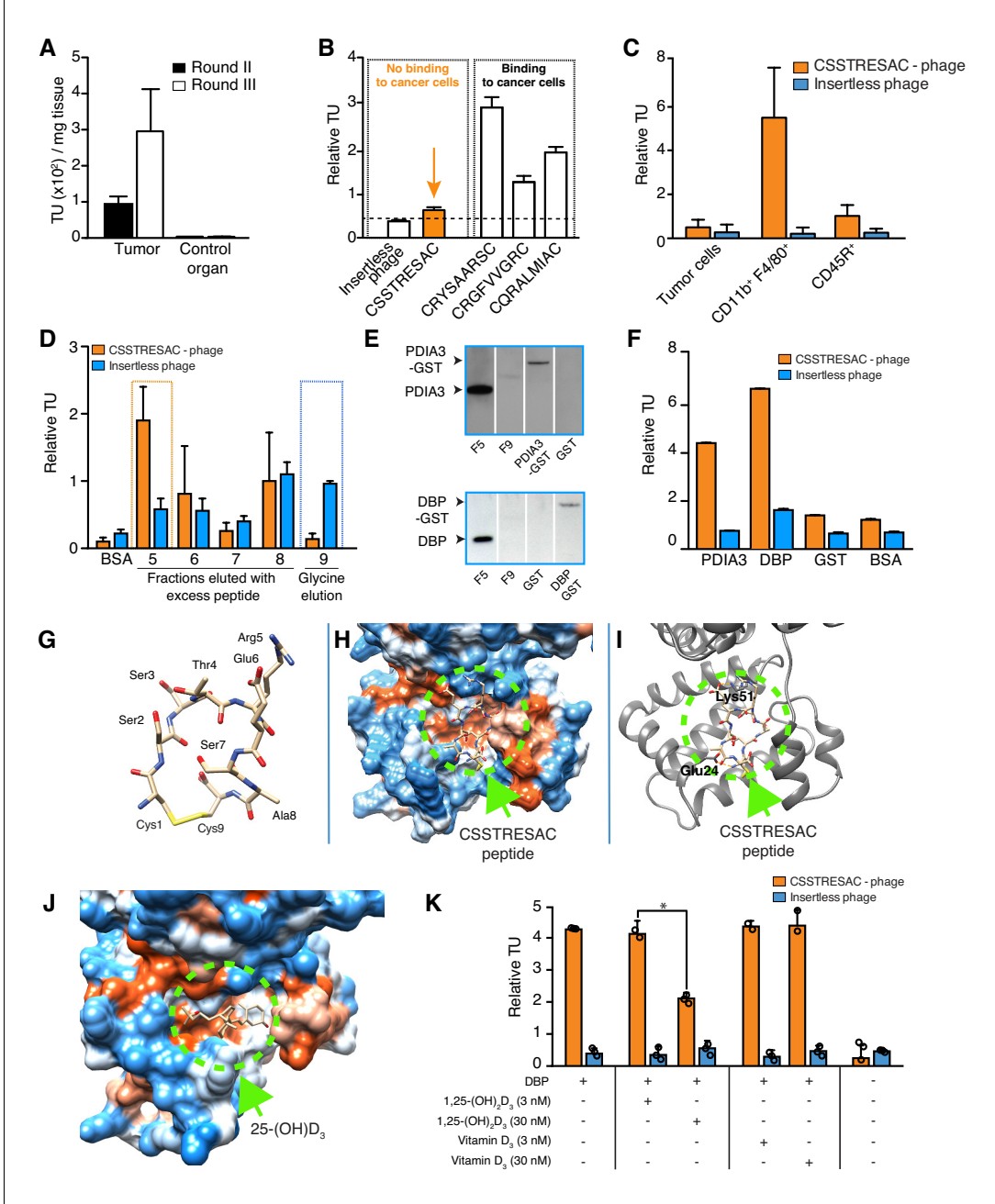

**Figure 1.** Combinatorial targeting of the tumor cellular microenvironment in a mouse model of TNBC. (A) A random phage display peptide library displaying CX7C inserts (C, cysteine; X any seven residues) was used in vivo to select peptides homing to the microenvironment of EF43.fgf4-derived mammary tumors. Three sequential rounds of selection resulted in a pool of targeted phage particles with a 300-fold enrichment in the tumor, compared to a control organ (muscle). (B) Binding of individual phage clones to EF43.fgf4 cells was quantified by the counting of transducing units (TU) after host bacterial infection. (C) Binding of CSSTRESAC-phage to EF43.fgf4 tumor cells and non-malignant stromal cell subpopulations isolated from mCherry-expressing EF43.fgf4-derived mammary tumors. (D) Relative binding of the CSSTRESAC-phage or insertless control phage to fractions eluted from a CSSTRESAC-conjugated affinity purification column. BSA was used as negative control protein. (E) Immunoblottings developed with either anti-PDIA3 (top panel) or anti-DBP (lower panel) antibodies show the presence of both affinity-purified proteins in the experimental fraction F#5 but not in the negative control fraction F#9. Human recombinant PDIA3-GST and DBP-GST were used as control for antibody specificity. (F) Phage-binding assay confirms preferential binding of targeted CSSTRESAC-phage to the recombinant human DBP. GST and BSA were used as negative controls. (G) Predicted structure of CSSTRESAC peptide, including a 2.0 Å-disulfide bridge between Cys1 and Cys9, as visualized with UCSF Chimera. (H) Predicted binding conformation and orientation of CSSTRESAC relative to the crystal structure of DBP in a hydrophobicity surface view (PDB ID: 1KW2_A). Orange and blue represent hydrophobic and hydrophilic patches, respectively. (I) Key predicted non-hydrophobic interactions between CSSTRESAC and DBP (PDB ID: 1KW2_A), including a 2.9 Å-salt bridge between Cys1 and Glu24, a 2.9 Å-salt bridge between Glu6 and Lys51, and a 2.9 Å-hydrogen

*Figure 1 continued on next page*

*Figure 1 continued*

bond between Ala8 and Glu24. CSSTRESAC also blocks access to Tyr48 and Ser92 (Tyr32 and Ser76 in PDB ID: 1J78), which correspond to predicted key residues of DBP interaction with 1,25-$(OH)_2D_3$. (**J**) Crystal structure of 25-$(OH)D_3$ bound to DBP in a hydrophobicity surface view (PDB ID: 1J78). Orange and blue represent hydrophobic and hydrophilic patches, respectively. (**K**) Binding of CSSTRESAC-phage to DBP is inhibited by the active form of vitamin D [1,25-$(OH)_2D_3$], but not by its corresponding vitamin D3 precursor (* represents Student's t-test, p<0.05).

The online version of this article includes the following figure supplement(s) for figure 1:

**Figure supplement 1.** EF43.fgf4-derived tumor is a model of triple negative mammary cancer.
**Figure supplement 2.** Macrophages are a major component of EF43.fgf4 mammary tumors.
**Figure supplement 3.** The CSSTRESAC peptide targets breast cancer in various mouse models.
**Figure supplement 4.** CSSTRESAC-targeted liposomes do not cause toxicity in mice.

and F#9 (negative control fraction) were resolved by sodium dodecyl sulfate-polyacrylamide gel electrophoresis (SDS-PAGE), and differential protein bands were subjected to in tandem mass spectrometry fragmentation (LS-MS/MS) for protein identification (*Supplementary file 1*). Notably, immunoblotting of eluted fractions revealed the presence of two vitamin D-binding receptor candidates: protein disulfide-isomerase A3 (PDIA3; also known as glucose-regulated protein-58 kDa, GRP58; endoplasmic reticulum protein of 57 kDa, ERp57; and membrane-associated rapid response to steroid-binding, 1,25$D_3$-MARRS) (*Khanal and Nemere, 2007*; *Figure 1E*, top panel) and vitamin D-binding protein (DBP) (*Figure 1E*, bottom panel). In vitro binding assays to recombinant PDIA3 and DBP confirmed preferential binding of CSSTRESAC-phage relative to the negative control insertless phage (*Figure 1F*).

## CSSTRESAC mimics active vitamin D

PDIA3 and DBP both bind to vitamin D (*Christakos et al., 2016*), thereby suggesting that CSSTRESAC might be structurally similar to vitamin D. Thus, we applied computational molecular modeling to determine whether the peptide CSSTRESAC would show conformational similarities to vitamin D (*Figure 1G–J*). The structure of CSSTRESAC was modeled with a de novo peptide structure prediction tool (PEP-FOLD2) (*Shen et al., 2014*; *Figure 1G*). Next, Rosetta FlexPepDock (*Raveh et al., 2011*) was used to identify putative binding site(s) for CSSTRESAC on the surface of DBP. Because the 3D structure of the DBP/1,25-$(OH)_2D_3$ complex was not available when this work was performed, we used a 2.3 Å-resolution X-ray crystal structure of the unliganded form of human DBP (PDB ID: 1KW2_A) (*Otterbein et al., 2002*). To initiate the docking calculation, CSSTRESAC was pre-positioned in the vicinity of the known binding site for 25-$(OH)D_3$, [and likely 1,25-$(OH)_2D_3$ based on previous computational modeling], visualized in the 2.1 Å-resolution X-ray crystal structure of a liganded form of human DBP (PDB ID: 1J78) (*Verboven et al., 2002*). The molecule 25-$(OH)D_3$, also known as calcidiol, binds at the base of a deep, largely hydrophobic pocket on the surface of domain I of DBP (*Figure 1J*). The computed model of the DBP/CSSTRESAC complex revealed a potential binding site for CSSTRESAC at the opening of the hydrophobic pocket. The computed model suggests that the largely hydrophilic peptide interacts with two superficial residues adjacent to the hydrophobic pocket, including Glu24 and Lys51 (*Figure 1H,I*; *Verboven et al., 2002*). The outcome of the Rosetta FlexPepDock calculations suggests that although CSSTRESAC binds at a similar site on the surface of DBP as 1,25-$(OH)_2D_3$ and its metabolite calcidiol, it is unlikely to interact more tightly and should be competitively displaced by the natural ligands of the receptor protein (*Figure 1I,J*). Indeed, experimental binding of CSSTRESAC-phage to immobilized DBP was reduced (Student's t-test, p<0.05) by increasing amounts of 1,25-$(OH)_2D_3$ but not by the non-active precursor vitamin $D_3$ (*Figure 1K*), a biochemical finding consistent with the computational model.

## PDIA3 is a receptor of the CSSTRESAC peptide and a novel molecular marker of TAM

Despite the fact that binding of CSSTRESAC to DBP is strongly suggested by our structural modeling, DBP is a circulating serum protein and thus unlikely to function as an integral cell surface receptor. Therefore, we reasoned that the membrane-bound receptor candidate PDIA3 would likely be the cell surface receptor on TAM responsible for the binding of CSSTRESAC. To determine whether PDIA3 is present on the cell surface of TAM in TNBC, we co-stained CD11b+ TAM isolated from

EF43.*fgf4* tumors with antibodies against IL-10, IL-12, and PDIA3. Flow cytometry analysis showed robust expression of PDIA3 on the surface of CD11b$^+$IL-10$^{high}$IL-12$^{low}$ TAM (*Figure 2A*), identifying PDIA3 as a new cell membrane-associated candidate marker of M2-polarized macrophages. Consistently, EF43.*fgf4* cells isolated from tumors did not express PDIA3 (*Figure 2B*), in agreement with the lack of CSSTRESAC-phage binding to EF43.*fgf4* cells. Moreover, immunofluorescence staining of frozen breast tumor sections from tumor-bearing mice receiving CSSTRESAC-phage iv suggested co-localization between PDIA3 and CD68, a well-established cell surface marker of macrophages (*Figure 2C,D*). Finally, administration of an anti-PDIA3 antibody into EF43.*fgf4* tumor-bearing mice confirmed accessibility of PDIA3 through the systemic circulation (*Ozawa et al., 2008*; *Figure 2—figure supplement 1A*). Notably, extracellular expression of PDIA3 was largely restricted to resident macrophage in tumors, while control tissues showed minimal cell surface staining. The macrophage marker F4/80 served as an additional positive control (*Figure 2—figure supplement 1B*).

## CSSTRESAC mimics active vitamin D, binds to DBP and mediates activation of PDIA3 on the surface of TAM

To gain insight into the biological mechanisms associated with this newly discovered ligand-receptor system, we next evaluated whether the predicted interactions between CSSTRESAC and PDIA3 on the surface of TAM would have functional consequences. We isolated CD11b$^+$F4/80$^+$ TAM from EF43.*fgf4* mammary tumors, established them in culture (>99% purity by FACS), and tested cytokine production as a surrogate for immunoregulatory responses upon treatment (*Figure 2E–G*). Cytokines were measured by real-time quantitative PCR after RNA extraction from cultured CD11b$^+$F4/80$^+$ TAM exposed to soluble CSSTRESAC. Untreated cultured CD11b$^+$F4/80$^+$ TAM served as negative controls. Treatment of CD11b$^+$F4/80$^+$ TAM with the soluble CSSTRESAC peptide induced a marked (on average ~40 fold) increase in gene expression of the pro-inflammatory cytokines IL-1β, TNF-α, and IL-6 (*Figure 2F,G*). In contrast, there was much lower increases in gene expression of the anti-inflammatory cytokines TGF-β1, TGF-β2, IL-10, and arginase-1 (*Figure 2E–G*) with IL-4 and IL-13 being undetectable. iNOS$_2$ (~20-fold) and the cytokine IL-23 (~10-fold) were also substantially increased upon exposure to CSSTRESAC. IL-18, IL-12, and INFγ showed modest increases or were detected only at background levels (*Figure 2F,G*; *Figure 2—figure supplement 1C*). This cellular response was abrogated when CSSTRESA-treated CD11b$^+$F4/80$^+$ TAM were co-treated with 1,25-(OH)$_2$D$_3$ (*Figure 2F,G*), verifying that it was specifically caused by the binding of the CSSTRESAC peptide. Thus, binding of CSSTRESAC directly to TAM may alter the local antitumor immune response through changes in cytokine production.

## Targeted ablation of PDIA3-expressing TAM affects tumor growth

We next investigated the biological significance and potential therapeutic effects of CSSTRESAC in the EF43.*fgf4* tumor model (*Figure 3A*). Mice bearing size-matched EF43.*fgf4* tumors were treated iv with soluble CSSTRESAC peptide, unrelated control peptide, or vehicle. A significant delay in tumor growth of mice treated with CSSTRESAC was observed as soon as one-week post initiation of treatment, compared to tumors of mice receiving an unrelated control peptide or vehicle alone (*Figure 3A*, t-test, p<0.001). FACS analysis of CD11b$^+$F4/80$^+$ TAM showed a reduction in the number of CD11b$^+$IL10$^{high}$IL12$^{low}$PDIA3-expressing TAM in tumors from mice treated with soluble CSSTRESAC peptide as compared to the negative control groups (*Figure 3B* and *Figure 3—figure supplement 1A*). Immunohistochemistry staining of representative tumor sections further demonstrated a reduction of the macrophage population in tumors treated with soluble CSSTRESAC (*Figure 3—figure supplement 1B*). Thus, treatment of tumors with the soluble CSSTRESAC peptide inhibited tumor growth and altered the TAM population in tumors, which supports it as a potential antitumor drug lead candidate.

As an additional medical application, we also analyzed the use of CSSTRESAC as a theranostic ligand for targeting transgenes directly to tumors in preclinical settings. We engineered adeno-associated/phage (AAVP) (*Dobroff et al., 2016*; *Ferrara et al., 2016*; *Hajitou et al., 2006*; *Smith et al., 2016*; *Staquicini et al., 2011*) constructs carrying the *Herpes simplex virus thymidine kinase* (*HSVtk*) gene to enable targeted suicide therapy upon combination with the pro-drug ganciclovir (GCV) (*Dobroff et al., 2016*; *Ferrara et al., 2016*; *Hajitou et al., 2006*; *Smith et al., 2016*; *Staquicini et al., 2011*; *Tjuvajev et al., 1998*). CSSTRESAC-AAVP-*HSVtk* or control AAVP lacking

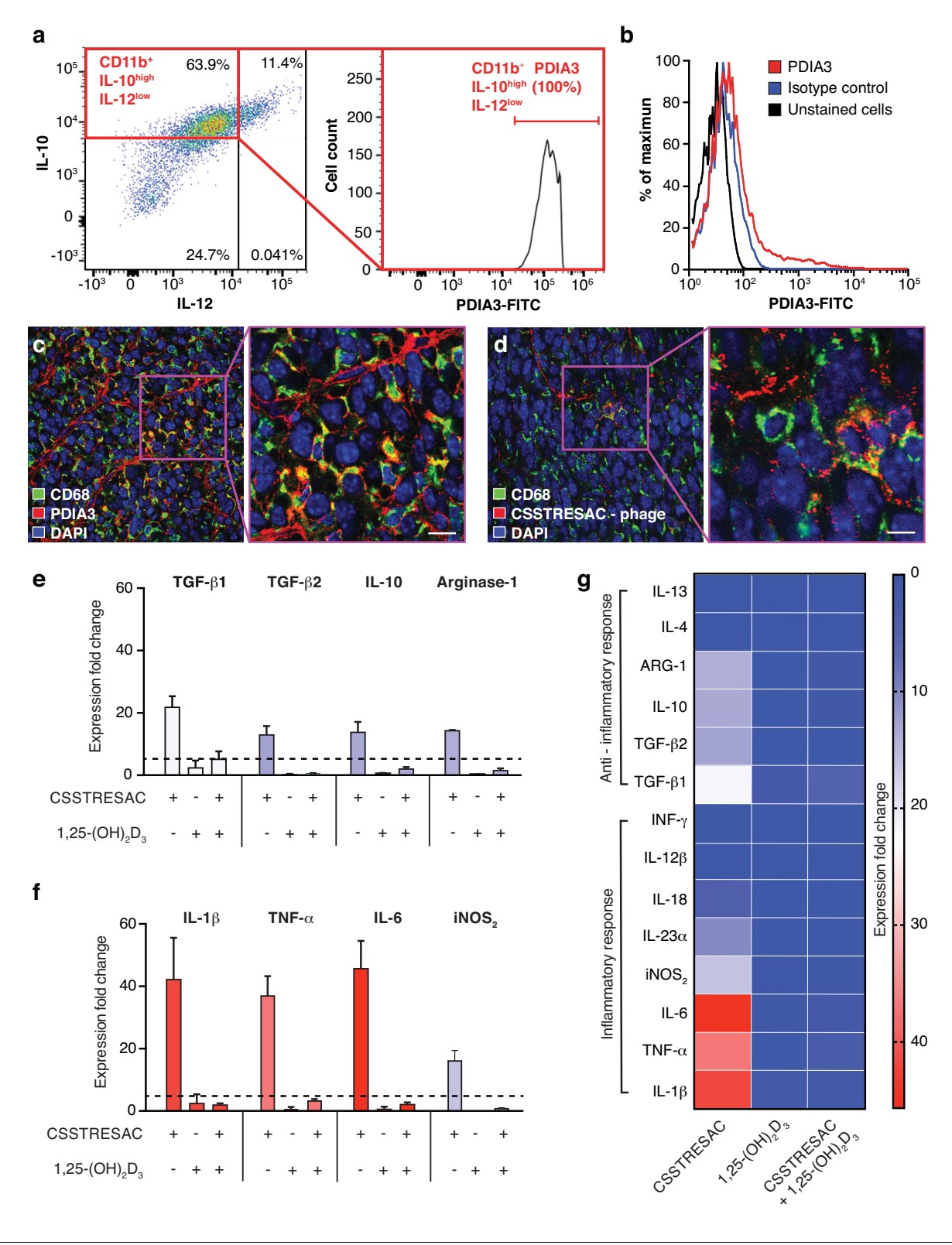

**Figure 2.** PDIA3 is present on the surface of TAM. (A) FACS analysis of total TAM isolated from EF43.fgf4-derived mammary tumors shows high levels of PDIA3 expression in a subpopulation of F4/80+CD11b+IL10highIL12low TAM. (B) EF43.fgf4 cells do not express detectable levels of PDIA3 on their surface. (C-D) PDIA3 expression in TAM and co-localization with the pan-macrophage marker CD68 as detected by immunofluorescence of tumor tissue sections from tumor-bearing mice administered iv with anti-PDIA3 antibody (C) or CSSTRESAC-phage (D). (E-G) Purified TAM from EF43.fgf4 mammary

*Figure 2 continued on next page*

*Figure 2 continued*

tumors were established in culture and treated with either the soluble CSSTRESAC peptide, 1,25-(OH)$_2$D$_3$, or both. Controls included untreated cells, and cells treated with vehicle. Expression of anti-inflammatory (**E** and **G**) or pro-inflammatory (**F** and **G**) cytokines in CD11b$^+$F4/80$^+$ TAM was assessed by quantitative real-time PCR. Graphics represent expression fold-change relative to control cells.

The online version of this article includes the following figure supplement(s) for figure 2:

**Figure supplement 1.** PDIA3 is accessible through the systemic circulation.

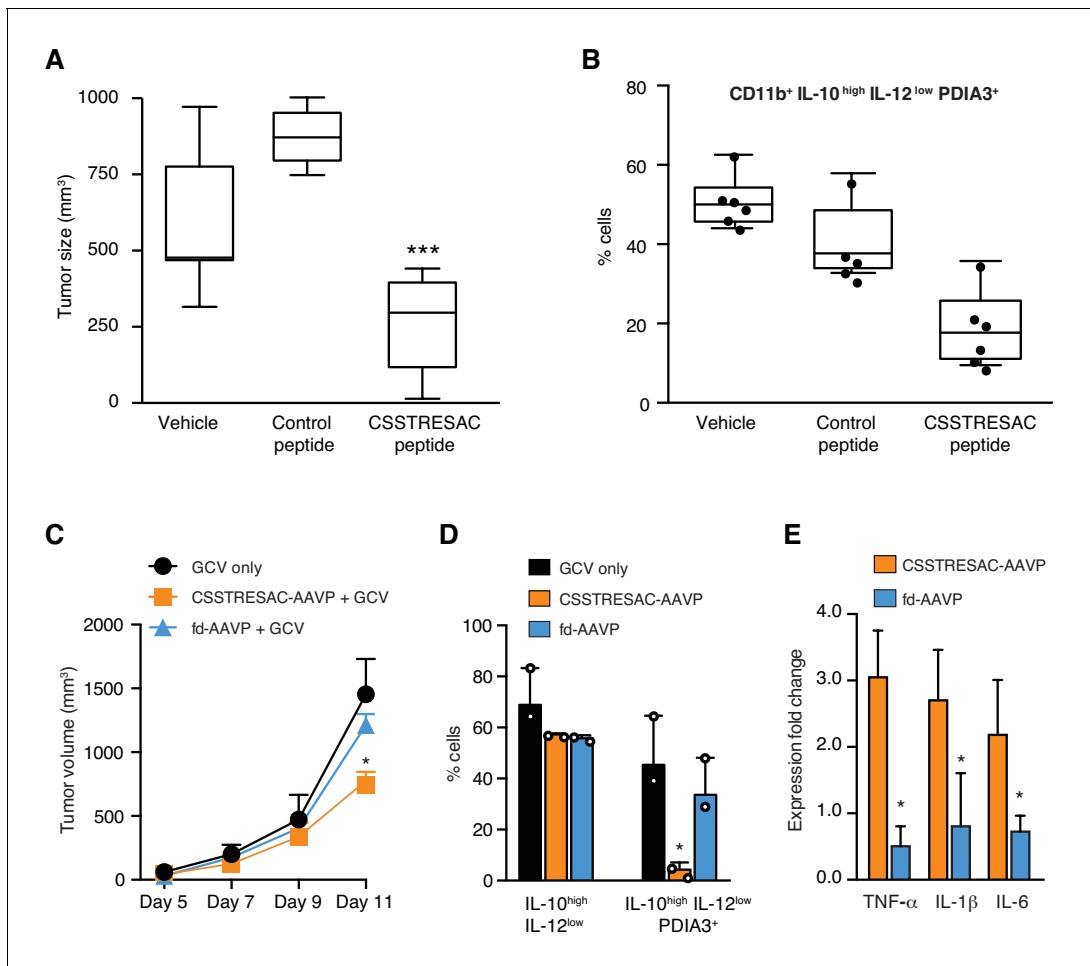

**Figure 3.** Targeted therapy delays growth of EF43.fgf4-derived mammary tumors. (**A**) Therapeutic effect of systemic treatment of EF43.fgf4 tumor-bearing mice with soluble CSSTRESAC peptide (n = 10 each experimental cohort, details in Materials and methods). An unrelated control peptide and vehicle served as negative controls. Tumor sizes were measured by digital caliper 1 week after treatment initiation, and every other day afterwards. *** represents p<0.001. (**B**) Treatment of tumor-bearing mice with CSSTRESAC reduces the number of PDIA3-expressing TAM (F4/80$^+$CD11b$^+$IL-10$^{high}$IL-12$^{low}$PDIA3$^+$). The TAM population is represented as percentage of total non-malignant cells, as determined by flow cytometry. (**C**) Gene therapy with CSSTRESAC-AAVP-*HSVtk* plus GCV delays tumor growth. Mice cohorts with size-matched EF43.fgf4 mammary tumors received a single systemic iv administration of targeted CSSTRESAC-AAVP-*HSVtk* (5 × 10$^{10}$ TU) or control fd-AAVP-*HSVtk*. Mice received daily doses of GCV (80 mg/kg/day) starting at day 7 post AAVP-*HSVtk* administration until the end of the experiment. * represents p<0.05. (**D**) Flow cytometry confirms reduction of F4/80$^+$CD11b$^+$IL-10$^{high}$IL-12$^{low}$PDIA3$^+$ TAM in tumors from CSSTRESAC-AAVP-*HSVtk*-treated mice. (**E**) Cytokine production by macrophages from tumors of mice treated with CSSTRESAC-AAVP-*HSVtk* or control groups. * represents p<0.05. Results are reported as expression fold-change relative to control group (set to 1).

The online version of this article includes the following figure supplement(s) for figure 3:

**Figure supplement 1.** CSSTRESAC peptide targets macrophage in vivo.

**Figure supplement 2.** Heat-map representing a more extensive cytokine profile of F4/80$^+$CD11b$^+$IL-10$^{high}$IL-12$^{low}$PDIA3$^+$ TAM isolated from tumors of treated and control groups.

the targeting peptide (fd-AAVP-*HSVtk*) were delivered to cohorts of size-matched EF43.*fgf4* tumor-bearing mice. Animals treated with vehicle were used as controls (n = 10, each cohort). All cohorts received GCV. By the end of the experiment, the sizes of tumors in mice that received CSSTRESAC-AAVP-*HSVtk* were significantly smaller than that of mice receiving control fd-AAVP-*HSVtk* or vehicle alone (*Figure 3C*, t-test, p<0.001). Moreover, macrophage quantification showed a reduction in the number of F480+CD11b+IL10highIL12lowPDIA3-expressing TAM (*Figure 3D*) accompanied by a shift in the cytokine profile toward an inflammatory response in the tumor microenvironment (*Figure 3E* and *Figure 3—figure supplement 2*).

The preclinical efficacies of soluble CSSTRESAC peptide and of CSSTRESAC-AAVP-*HSVtk* were further investigated in silico. We have conceived a mathematical model of tumor growth and treatment efficiency to predict response in breast cancer patients. This mechanistic model was formulated as a system of ordinary differential equations based on our prior work on modeling cancer response to various forms of drug treatment (*Brocato et al., 2018*; *Brocato et al., 2019*; *Dogra et al., 2018*; *Goel et al., 2019*; *Wang et al., 2016*). The model accounts for two primary opposing processes: tumor cell growth and death caused by the CSSTRESAC peptide, while also allowing for competitive antagonism exhibited by 1,25-$(OH)_2D_3$ in serum. To model tumor growth delay in gene therapy experiments, an extra death rate term was introduced that characterizes death due to GCV activated through *HSVtk* (equations are described in Materials and methods). Model predictions corroborated with experimental data from mouse models (Pearson correlation coefficient $R = 0.998$, p = 0.001) and were used to simulate a clinical trial for treatment of breast cancer patients with soluble CSSTRESAC (*Figure 4* and *Figure 4—figure supplement 1*). To evaluate the importance of a possible competitive binding between 1,25-$(OH)_2D_3$ and soluble CSSTRESAC in the serum, the dissociation constant $K_d$ was perturbed by ± 20% of the reference parameter value. As such, an increase in $K_d$ would reflect the competitive binding of the antagonist 1,25-$(OH)_2D_3$, where the dissociation of CSSTRESAC from PDIA3 on the cell surface increases and the antitumor effects of CSSTRESAC decreases. Similarly, a reduced $K_d$ would reflect a stronger binding between CCSTRESAC and PDIA3 with the consequent inhibition of tumor growth (*Figure 1—figure supplement 1A*). We have also considered a hypothetical experiment in patients where a constant rate of i.v. infusion of the soluble CSSTRESAC peptide was compared to the efficacy of a unit i.v. bolus. Our mathematical model predicted that infusion of CCSTRESAC would result in ~400 mm$^3$ greater reduction in tumor volume compared to bolus (*Figure 4A* and *Figure 4—figure supplement 1B*). Our proposed working hypothesis shows that the CSSTRESAC-DBP complex specifically binds to PDIA3 and elicits functional changes in PDIA3-expressing TAM within the tumor microenvironment. Such biochemical and cellular alterations may in turn result in an inflammatory local response potentially mediated by IL-6, IL-1β, and TNF-α, and inhibition of tumor growth (*Figure 4B*).

Lastly, we searched a publicly available single-cell transcriptome dataset of breast cancer and immune-infiltrating cells containing data from TNBC patients for PDIA3-expressing TAM. Transcripts per million reads (TPM), single-cell (sc)RNA-seq and sample information were obtained from the Gene Expression Omnibus (GEO) repository (accession #GSE75688) (*Chung et al., 2017*); an initial gene set variation (GSVA) analysis extracted single cells (n = 35) displaying gene expression pathways of infiltrating macrophages. Expression of the *PDIA3* gene in these cells was deemed high, medium, or low, and it was clustered/plotted relative to the expression of established markers of immune suppression and M2-polarized macrophages (*IL10*, *TGFB1*, *CD274*, *PDCD1LG2*, *CD68*, *CD163*, *ITGAM*, *CXCL2*, and *MS4A6A*). Markers of angiogenesis and/or disease progression (*PLAUR*, *IL8*, *VEGFA*, and *MMP9*) were also included (*Garrido-Castro et al., 2019*; *DeNardo and Ruffell, 2019*; *Lim et al., 2018*; *Wagner et al., 2019*). An unsupervised clustering analysis (*Figure 5*) showed that high levels of *PDIA3* expression in TAM clustered positively with markers of M2-polarized TAM as well as poor prognosis indicators and genes associated with immune suppression. These genomic results support the presence of PDIA3-expressing TAM in human TNBC, and suggest that these preclinical findings may be clinically meaningful.

## Discussion

We report that PDIA3 is a functional receptor expressed on the cell surface of the M2-like class of TAM in TNBC. We show that PDIA3, an established vitamin D-interacting protein, has immunoregulatory functions as the TAM cell surface receptor for the peptide CSSTRESAC, with clear effects in

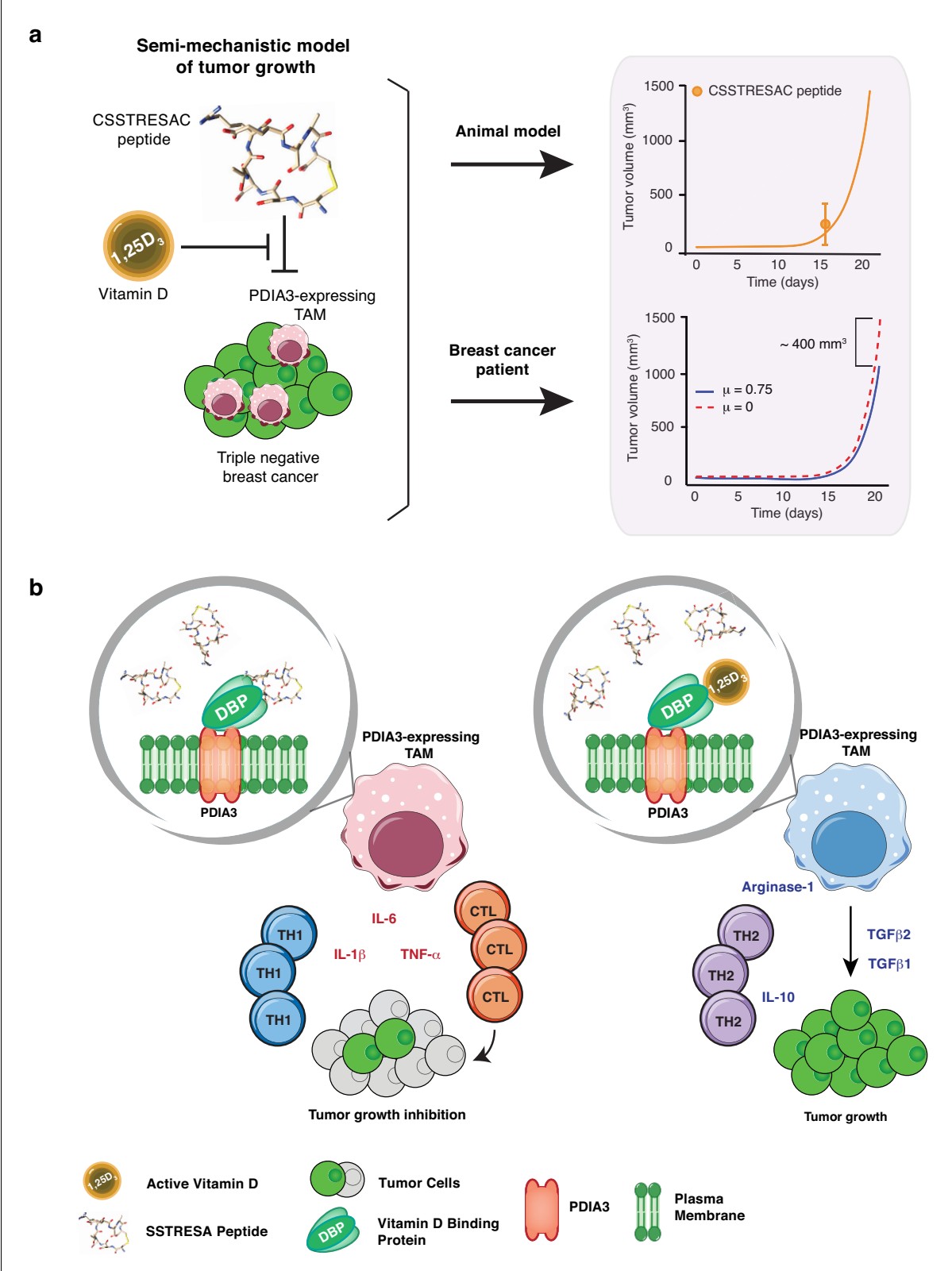

**Figure 4.** Mechanistic mathematical model of tumor growth inhibition upon treatment with soluble CSSTRESAC and competitive antagonism by 1,25-(OH)$_2$D$_3$. (**A**) System interactions captured by a mechanistic mathematical model. Upper panel shows the non-linear regression of the tumor growth model upon treatment of tumor-bearing mice with soluble CSSTRESAC. Error bar represents mean ± standard deviation (S.D.) of the data shown in *Figure 3A*. Lower panel shows the projected temporal evolution of the tumor volume without infusion (μ = 0) and with infusion (μ = 0.75) in a simulated

*Figure 4 continued on next page*

*Figure 4 continued*

human clinical trial. (**B**) A schematic representation of the working hypothesis. The complex CSSTRESAC-DBP binds PDIA3 and eliminates PDIA3-expressing TAM from the tumor microenvironment (through an unknown mechanism), resulting in a pro-inflammatory local response and inhibition of tumor growth. Because 1,25-(OH)$_2$D$_3$ may compete out the effects of CSSTRESAC, binding to PDIA3-expressing TAM in the presence of 1,25-(OH)$_2$D$_3$ may be abrogated, and tumor cells can continue to grow.

The online version of this article includes the following figure supplement(s) for figure 4:

**Figure supplement 1.** Mathematical modeling of CSSTRESAC peptide distribution in tumor-bearing mice.

preclinical non-TNBC and TNBC mouse models and, at least potentially, in TNBC patients. The effects of soluble CSSTRESAC and CSSTRESAC-AAVP-*HSVtk* in the local and systemic immune responses in murine models of breast cancer also suggest that combination therapy with immuno-modulators may increase the therapeutic response against highly inflammatory tumors. In particular, TNBCs are more likely to respond to immunotherapy due to higher numbers of tumor-infiltrating lymphocytes, higher levels of PD-L1 expression in both tumor cells and immunce cells as well as a higher mutational burden and the consequent rise in tumor-specific neo-antigens. Therefore, immu-nomodulation of the local and/or systemic responses with immune checkpoint inhibitors could—at least in theory—be amplified by CSSTRESAC-mediated immunoregulatory functions in breast can-cer, as well as other TAM-infiltrated cancers, and might perhaps become a medically meaningful translational strategy. In this setting, CSSTRESAC could also be considered as a new non-steroidal vitamin D analogue prototype for drug lead-optimization, with applications that may include other diseases (malignant or non-malignant) with an inflammatory component.

Finally, we introduce a ligand-directed AAVP-*HSVtk* platform for theranostics based on cell sur-face targeting of PDIA3 along with a mathematical model that reproduces the experimental dataset and estimates CSSTRESAC treatment outcomes in breast cancer. These observations in vitro, in

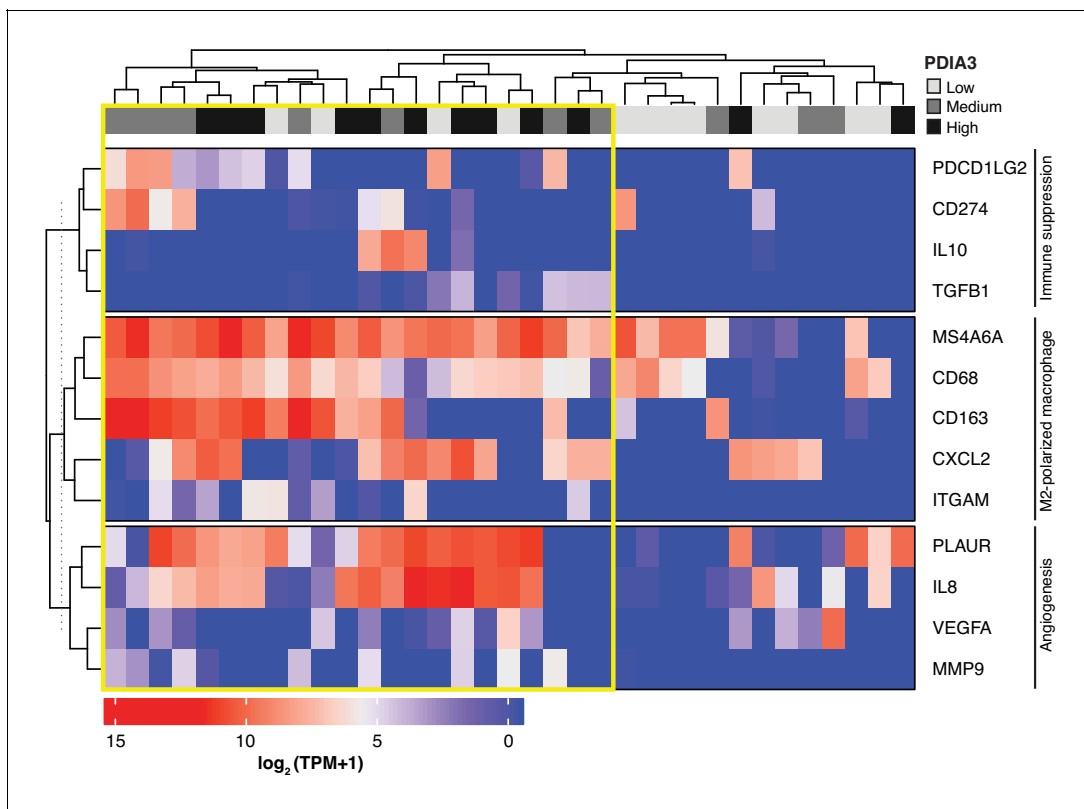

**Figure 5.** Heat-map of PDIA3 gene expression in pre-defined myeloid cells from human TNBC. The heat map shows a strong association with the expression of genes characteristic of M2-polarized macrophage, markers of immunosuppression and angiogenesis (i.e. poor prognosis). The yellow box highlights cells with the highest expression of PDIA3.

mouse mammary tumor models, plus an initial in silico analysis of cells from TNBC patients, support an unrecognized regulatory role of PDIA3-expressing TAM in the tumor immune response. Finally, one should note that our mathematical model shows that native competing serum 1,25-$(OH)_2D_3$ is unlikely to influence the binding of CSSTRESAC to its target on TAM. Notably, the Human Protein Atlas shows cytoplasmic expression of PDIA3 in human breast cancer cells. Thus, expression of PDIA3 on the surface of cancer cells, and potential effects of the direct binding of CSSTRESAC to breast cancer cells warrants further investigation and, if confirmed, might have translational implications in the setting of TNBC, and other human tumors or even non-malignant disorders with a inflammatory component. Similarly, drug interactions caused by prolonged exposure to CSSTRESAC in the presence of steroids—which are often used in cancer patients—might confound the investigational use of CSSTRESAC-based therapies and should be carefully considered.

## Materials and methods

### Antibodies and recombinant proteins

Anti-PDIA3, anti-IL-10, anti-IL-12, and anti-F4/80 were purchased from BD Pharmingen and were used in flow cytometry and immunofluorescence. Immunoblottings were performed with antibodies purchased from Sigma (Glutathione S-transferase, PDIA3 and DBP), Abcam (ER), Cell Signaling (PgR), and R and D Systems (HER2). Taqman assays for real-time PCR quantification of cytokines were purchased from Applied Biosystems. Recombinant proteins (DBP and PDIA3), cholecalciferol and calcitriol were all acquired from Abcam. Fluorescence-conjugated secondary antibodies were purchased from Jackson Immunoresearch. Peptides were custom synthesized by PolyPeptide Laboratories to our specifications (>95% purity).

### Cells lines and tissue culture

Mouse mammary EF43.*fgf4* cells (*Adams et al., 1987*; *Hajitou et al., 1998*) were maintained in Dulbecco's modified Eagle's medium (DMEM) supplemented with 10% fetal bovine serum (FBS), 5 ng/ml mouse epithelial growth factor (EGF), 1 μg/ml bovine insulin, and antibiotics. MDA-MB-231 cells (*Cailleau et al., 1974*) were obtained from the American Type Culture Collection (ATCC) and were grown as a monolayer in RPMI 1640 medium supplemented with 8.25% FBS. Cells were maintained at 37°C and 5% $CO_2$. All cells were routinely tested for the presence of mycoplasm. ATCC garantees the identity of purchased cells.

### Animals and experimental tumor models

Eight-week-old female nude (nu/nu) mice and immunocompetent BALB/c mice were housed in the animal facilities of the University of Texas M.D. Anderson Cancer Center # 11-99-09935 and Rutgers University New Jersey Medical School (PROTO201800055), all in the USA. Polyoma middle T transgenic (PyMT) mice were maintained at the University Medical Center Hamburg-Eppendorf, Germany. All animal procedures were reviewed and approved by the corresponding Institutional Animal Care and Use Committee (IACUC)-equivalent at each institution.

Both human MDA-MB-231 cells and mouse EF43.*fgf4* cells were implanted in the mammary fat pads of nude and immunocompetent BALB/c mice, respectively. Tumor-bearing mice were sorted into experimental size-matched cohorts when established tumors reached ~200 $mm^3$. These procedures were conducted in accordance with the Guide for the Care and Use of Laboratory Animals published by the U.S. National Institutes of Health (NIH Publication #85–23, revised 1996) approved by the local ethics review board.

PyMT [strain FVB/N-TgN (MMTVPyVT)634-Mul] (*Michelfelder et al., 2009*) were obtained from Jackson Laboratory (Bar Harbor, ME, USA). All procedures involving PyMT mice were conducted in accordance with the German Animal Protection Code and approval was granted by the local ethics review board (Hamburg, Germany). PyMT transgenic mice genotyping was performed through blood samples collected from the retrobulbar venous plexus under anesthesia (2% isoflurane, 98% oxygen), as described (*Michelfelder et al., 2009*).

Treatment of tumor-bearing mice with a single-dose of CSSTRESAC-AAVP-HSV*tk* or control fd-AAVP-HSV*tk* (5 × $10^{10}$ TU per mouse) was followed by daily intraperitoneal (ip) administrations of

GCV at 80 mg/kg/day. Tumor sizes were measured every-other day with a digital caliper and plotted as tumor volume ($mm^3$).

## Phage display methodology

The Biopanning and Rapid Analysis of Selective Interactive Ligands (BRASIL) methodology (*Giordano et al., 2001*) was used to test binding of phage to cultured cells. For phage binding to the candidate receptors PDIA3 and DBP, individual microtiter wells of 96-well plates were coated overnight (ON) with 1 μg/ml of recombinant proteins, followed by blocking with BSA and incubation with $10^9$ TU of insertless phage or CSSTRESAC-phage for 1 hr at room temperature (RT). GST and BSA were used as control proteins. Bound phage were recovered by log-phase infection of host bacteria (200 μl *E. coli* K91Kan). Competitive binding of CSSTRESAC-phage and 1,25-$(OH)_2D_3$ to DBP was performed by using the same experimental protocol. Competition was performed in wells pre-incubated with 3 nM or 30 nM of either 1,25-$(OH)_2D_3$ or cholecalciferol.

Combinatorial phage display selections in vivo in tumor-bearing mice were performed as described (*Dobroff et al., 2016*; *Ferrara et al., 2016*; *Hajitou et al., 2006*; *Smith et al., 2016*; *Staquicini et al., 2011*; *Arap et al., 1998*; *Marchiò et al., 2004*; *Pasqualini and Ruoslahti, 1996*). In brief, animals received $10^9$ TU iv of an unselected phage display random peptide library (displaying the insert CX$_7$C). Tumors and control organs were collected after 24 hr of systemic circulation. For homing of individual phage clones in vivo, tumor-bearing mice were deeply anesthetized with 1–2% isofluorane and received $10^9$ TU of targeted phage or insertless control phage, both administered iv side-by-side. Phage particles were recovered from tissue samples by bacterial infection and processed as described (*Dobroff et al., 2016*; *Ferrara et al., 2016*; *Hajitou et al., 2006*; *Smith et al., 2016*; *Staquicini et al., 2011*; *Arap et al., 1998*; *Marchiò et al., 2004*; *Pasqualini and Ruoslahti, 1996*).

## Peptide affinity chromatography

Receptor candidates were isolated by using an affinity chromatography CarboxyLink column (ThermoFisher Scientific) conjugated with the synthetic CSSTRESAC peptide. Protein extracts (10 mg/purification) were added to peptide conjugated columns and incubated ON at 4°C under constant gentle agitation. After extensive washes, bound proteins were eluted with an excess of soluble CSSTRESAC peptide followed by elution in low pH glycine buffer. Contaminants including detergents, salts, lipids, phenolics, and nucleic acids were removed through a 2-D clean-up kit from GE Healthcare Life Sciences. Proteins were re-suspended in rehydration buffer (8 M urea, 2% CHAPS, 40 mM DTT, 0.5% IPG buffer, 0.002% bromophenol blue) and 2-D gel electrophoresis was performed by using the ZOOM IPGRunner System (Life Technologies). The final gel was stained with SYPRO Ruby Protein Gel Stain (Life Technologies) and imaged in a 300 nm ultraviolet transilluminator. Unique bands were excised from the SDS gels and digested with trypsin. LC-MS/MS analysis was performed at the Proteomics Core Facility of the University of Texas M.D. Anderson Cancer Center.

To test purified fractions for the presence of candidate receptors, control and experimental fractions were immobilized on individual microtiter wells of 96-well plates ON at 4°C. Wells were blocked with phosphate-buffered saline (PBS) containing 3% BSA for 1 hr at RT and incubated with $10^9$ TU of insertless phage or CSSTRESAC-phage. After extensive washing with PBS, bound phage particles were recovered by infection of host bacteria.

## Peptide structure prediction and docking

The peptide sequence of CSSTRESAC was entered into PEP-FOLD2 (*Shen et al., 2014*) with a designated disulfide bridge between Cys1 and Cys9 (to ensure the cyclic peptide configuration) and 100- and 200-run simulations were applied. The best-fit model containing a disulfide bridge between Cys1 and Cys9 based on sOPEP energy (i.e. the negative value with greatest absolute value) was selected as the structure for further experimentation. By using the UCSF Chimera (*Pettersen et al., 2004*), a PDB file with CSSTRESAC positioned adjacent to human DBP (PDB ID: 1KW2_A) (*Otterbein et al., 2002*) in roughly the same location as 25-$(OH)D_3$ in its complex with human DBP (PDB ID: 1J78) (*Verboven et al., 2002*) was generated and inputted into Rosetta FlexPepDock (*Raveh et al., 2011*). The top generated model according to energy scoring, a revised version of

Rosetta full-atom and coarse-grained energy functions, with CSSTRESAC bound to the same binding pocket as 25-(OH)D$_3$ was selected for analysis. Interacting residues of CSSTRESAC and DBP were analyzed via UCSF Chimera (*Pettersen et al., 2004*).

## Immunohistochemistry, immunofluorescence, and flow cytometry

For immunohistochemistry and immunoflorescence, the anti-PDIA3 antibody was administered iv into the tail vein of EF43.*fgf4* tumor-bearing BALB/c mice. After 5 min, the mice were killed and perfused through the heart. Tumors and control organs were collected and either quickly-frozen in liquid nitrogen or preservative-fixed, and paraffin-embedded (*Dobroff et al., 2016*; *Ferrara et al., 2016*; *Hajitou et al., 2006*; *Smith et al., 2016*; *Staquicini et al., 2011*; *Arap et al., 1998*; *Marchiò et al., 2004*; *Pasqualini and Ruoslahti, 1996*). The presence of the anti-PDIA3 antibody in tissue sections was verified by detection with a secondary antibody conjugated to horseradish peroxidase (HRP) or were stained for the presence of macrophages with an anti-CD68 antibody conjugated to FITC. For flow cytometry, whole EF43.*fgf4* tumors were dissected out from tumor-bearing BALB/c mice and single-cell suspensions were prepared by tumor mincing. The single-cell suspension was washed with PBS containing 5% FBS and 0.01% NaN$_3$. Cell suspensions were aliquoted into 12 × 75 mm flow cytometry tubes as 5 × 10$^5$ cells per tube and ice-cold incubated for 15 min with an Fc receptor blocking agent, followed by antibodies against PDIA3, F4/80, IL-10, and IL-12. Cells were incubated on ice for 30 min, followed by washes and secondary antibodies.

## Quantitative real-time PCR

Three sets of total RNA (RNeasy Mini Kit, Qiagen) were independently isolated from cultured macrophages, or fresh macrophages isolated directly from tumors. DNA synthesis was performed with the GoScript Reverse Transcription System (Promega) by using oligodT for reverse transcription. Gene expression was analyzed with the use of Taqman probes (Applied Biosystems) in a 7500 Fast Real-Time PCR System instrument (Applied Biosystems) and three sets of endogenous control genes: 18S and GAPDH and GUSB1.

## Macrophage isolation and tissue culture

TAM were obtained directly from EF43.*fgf4* tumors. Tissue digestion was performed in collagenase A in serum-free DMEM (1 mg/mL) for 20 min at 37°C, followed by filtering through 70 µm nylon cell strainers and centrifugation. Macrophages were enriched by magnetic bead separation of CD11b-positive cells (Miltenyi Biotec) and either used for RNA extraction or cultured in 6-well plates containing DMEM (Gibco) supplemented with 20% FBS (Sigma) and 50 ng/ml of M-CSF (R and D Systems). A homogeneous population of adherent macrophages (namely,>99% CD11b$^+$F480$^+$) was obtained after 7 days in culture.

## Preparation and characterization of liposomes

Cationic lipids DOTAP, DOPE, 1,2-Dioleoyl-sn-Glycero-3-Phosphoethanolamine-N-[4-(p-maleimido-phenyl)butyramide] (DOPE-MPB), and DOPE-rhodamine B were purchased from Avanti Polar Lipids. Gadolinium (Gd)-BOA was commercially obtained (Gateway Chemical Technology). Liposomes were prepared by lipid hydration. DOTAP:DOPE:DOPE-MPB (1:0.95:0.05, mol/mol/mol) and were dissolved in chloroform in a round-bottom flask. DOPE-rhodamine B at a concentration of 0.2 mol was included for visualization of liposomes by high-resolution fluorescence microscopy. The total concentration of lipids was determined after extrusion by using the erythrosine method and found to be 10 mg/ml. Gd-BOA (25 mol %) was added to the base formulation in place of different molar fraction of DOTAP. The solvent was removed by evaporation by using nitrogen flow and the lipid film was hydrated by 5% dextrose solution. After hydration, the liposomes were left under an argon blanket at 4°C ON to allow annealing and 24 hr later the suspension of lipids was vortexed for 5 to 10 min, to allow liposome formation, and passed through a 200 nm pore size polycarbonate membrane through an extruder (Avestin Inc). The surface of liposomes was decorated with targeted and control peptides via maleimide chemistry. Fluorescein-labeled control peptide was synthesized by the Synthesis and Sequencing Facility of Johns Hopkins University School of Medicine. A total of 250 µg of either targeted or control peptide was added to the liposomal suspension and left for 24 hr at 4°C to allow covalent coupling. Subsequently, 300 µg of N-ethylmaleimide (NEM; Pierce) were added to

the liposomal suspension and kept for 2 hr at RT to block free sulfhydryl groups. Uncoupled peptide and excess of NEM were separated by using Sephadex G-100 size exclusion chromatography.

The hydrodynamic diameters and ζ-potential of liposomes without peptides, liposomes with targeted peptide, or control peptide were measured in 10 mM NaCl at 25°C in two independent experiments in a Malvern Zetasizer (Malvern Inc). Each measurement was repeated at least three times. The total concentration of Gd-BOA incorporated within the lipid bi-layer was determined with inductively coupled plasma mass spectrometry (ICP-MS from Perkin Elmer). The hydrodynamic diameter of liposomes with targeted or control peptides was in the range 150–200 nm with polydispersity index less than 0.2 nm. The presence of either targeted or control peptides on the surface of liposomes did not affect size distribution, and suspensions of liposomes were stable for several months at 4°C. Additional characterization of these liposomes revealed a surface charge of 27.5 ± 2.6 mV for uncoupled liposomes, 26.2 ± 0.9 mV for targeted liposomes, and 24.3 ± 1.2 mV for control liposomes.

## Magnetic resonance imaging and optical imaging

Cohorts of female nu/nu mice (n = 21) were inoculated in the mammary fat pad with $2 \times 10^6$ MDA-MB-231 cells suspended in 50 µl of Hanks balanced salt solution (ThermoFisher Scientific). Prior to tumor implantation, T1 relaxation times of the liposomal solutions were measured on a 4.7 T Bruker Biospec spectrometer horizontal bore magnet (Bruker BioSpin GmBH) with an inversion recovery sequence (repetition time [TR] 2000 milliseconds, number of averages [NA] 1 and 10 relaxation delays of 5, 10, 15, 20, 40, 60, 80, 100, 400, 800 ms). MRI studies were performed when tumor sizes reached ~300–350 mm$^3$. Multi-slice T1-weighted images were acquired with a multislice-spin echo (MSME) sequence (echo time [TE] 11.4 ms, TR 500 milliseconds, NA 2, field of view [FOV] 1.6 cm, matrix size 128 × 128, slice thickness 1 mm, from 6 to 8 slices). Quantitative T1 multi-slice maps with relaxation delays of 100, 500, 1000, and 7000 ms were obtained with TE 0.98 milliseconds, TR 500 ms, NA 8, FOV 1.6 cm, matrix size 128 × 128, slice thickness 1 mm with a modified SNAPSHOT FLASH sequence. The MRI scans were acquired before and at 3, 6, 24, 48, and 72 hr following iv administration of targeted or control liposomes. Images were processed by using customized analyses programs developed in Interactive Data Language (IDL; ITT Visual Information Solutions).

Biodistribution studies of Gd-BOA incorporated within lipid bi-layers of liposomes were performed on an 11.7 T wide-bore MR spectrometer (Bruker BioSpin GmBH) equipped with triple-axis gradients. T1 relaxation times of tumor, liver, kidney, spleen, lungs, intestine, heart, blood, and muscle (n = 3 each) were measured with an inversion recovery sequence (TR = 20 s, NA = 1 and 10 relaxation delays: 5, 10, 15, 20, 40, 60, 80, 100, 400, 800 ms for liver, spleen, and blood, and 40, 80, 100, 200, 400, 600, 1000, 5000, 8000, 10,000 milliseconds for triplicates of tumor, kidney, heart, lungs, and muscle). Biodistribution studies were also performed by using the fluorescent signal from rhodamine-labeled liposomes and FITC-labeled targeted or control peptide. Tumor-bearing mice (n = 3 in each group) received either targeted or control liposomes iv. Mice were killed at each time point and 1-mm-thick slices of tumor, liver, kidney, spleen, lungs, intestine, heart, and muscle were imaged in a Xenogen IVIS 200 optical imaging device (PerkinElmer).

To rule out hepatic toxicity associated to liposomal administration, alanine aminotransferase (ALT) and aspartate aminotransferase (AST) assay kits were purchased from Pointe Scientific Inc Levels of ALT and AST in mice serum were measured 48 hr post iv administration of targeted liposomes, control liposomes, or vehicle-only.

## Mathematical model of tumor growth and treatment efficiency

We developed a mechanistic model of tumor growth formulated as a system of ordinary differential equations based on our prior work on modeling cancer response to various forms of drug treatment (*Brocato et al., 2018*; *Brocato et al., 2019*; *Dogra et al., 2018*; *Goel et al., 2019*; *Wang et al., 2016*). The model accounts for two primary opposing processes: tumor cell growth and death caused by the CSSTRESAC peptide, while also allowing for competitive antagonism exhibited by 1,25-(OH)$_2$D$_3$ in serum. To model tumor growth delay in gene therapy experiments, an extra death rate term was introduced that characterizes death due to GCV activated through *HSVtk*. Specifically, the tumor proliferation rate (*G*) is characterized through a logistic equation, the death rate (*S*) due to the peptide is modeled as a Michaelis-Menten kinetics process, and the death rate (*N*) due to GCV

is modeled as a linear function of the concentration of GCV in plasma. Therefore, we obtain the following generic tumor growth model (equations 1 - 4), developed to capture changes in tumor volume ($V$) over time:

$$\frac{dV(t)}{dt} = G - S - N,$$ (1)

$$G = \sigma \cdot V(t) \cdot \left(1 - \frac{V(t)}{K}\right)$$ (2)

$$S = \frac{D_{\text{Pep}} \cdot C^{\text{P}}_{\text{Pep}}(t)}{C^{\text{P}}_{\text{Pep}}(t) + K_{\text{d}}}$$ (3)

$$N = \lambda \cdot C^{\text{P}}_{\text{GCV}}(t)$$ (4)

where $V_0$ is the initial volume of the tumor, $\sigma$ is the tumor growth rate constant, $K$ is the carrying capacity of the host, $C^{\text{P}}_{\text{Pep}}(t)$ is the plasma concentration of CSSTRESAC-AVVP-HSV$tk$, $C^{\text{P}}_{\text{GCV}}(t)$ is the plasma concentration of GCV, $D_{\text{Pep}}$ is the asymptotic death rate due to the peptide (indicative of potency), $K_{\text{d}}$ is the CSSTRESAC-DBP-PDIA3 complex dissociation constant—which is implicitly a function of the concentration of 1,25-(OH)$_2$D$_3$—and $\lambda$ is the proportionality constant between tumor volume and GCV concentration in plasma $C^{\text{P}}_{\text{GCV}}(t)$.

To estimate $C^{\text{P}}_{\text{Pep}}(t)$ and $C^{\text{P}}_{\text{GCV}}(t)$ for use in the tumor growth model, a one compartment pharmacokinetic (PK) model was employed. Given that the peptide was administered iv, we assumed a first-order renal clearance of the peptide, characterized by an excretion rate constant $k^{\text{Pep}}_{\text{ex}}$, such that:

$$\frac{dC^{\text{P}}_{\text{Pep}}(t)}{dt} = -k^{\text{Pep}}_{\text{ex}} C^{\text{P}}_{\text{Pep}}(t), \qquad C^{\text{P}}_{\text{Pep}}(0) = C_0$$ (5)

where $C_0$ is the initial plasma concentration of soluble CSSTRESAC.

Further, given that GCV was administered ip, in addition to first-order renal excretion (rate constant $k^{\text{GCV}}_{\text{ex}}$), a first-order absorption of GCV from the peritoneal cavity (rate constant $k^{\text{GCV}}_{\text{a}}$) into the bloodstream was also incorporated to model its plasma concentration $C^{\text{P}}_{\text{GCV}}(t)$. Hence, we obtain:

$$\frac{dC^{\text{P}}_{\text{GCV}}(t)}{dt} = k^{\text{GCV}}_{\text{a}} \cdot C^{\text{IP}}_{\text{GCV}}(t) - k^{\text{GCV}}_{\text{ex}} \cdot C^{\text{P}}_{\text{GCV}}(t)$$ (6)

$$\frac{dC^{\text{IP}}_{\text{GCV}}(t)}{dt} = -k^{\text{GCV}}_{\text{a}} \cdot C^{\text{IP}}_{\text{GCV}}(t)$$ (7)

$$\frac{dC^{\text{P}}_{\text{Pep}}(t)}{dt} = I(t) - k^{\text{Pep, sol}}_{\text{ex}} C^{\text{P}}_{\text{Pep}}(t), I(t) = \begin{cases} 0, & t \leq 5 \\ \mu \cdot k^{\text{Pep, sol}}_{\text{ex}}, & t > 5 \end{cases}$$ (8)

where $C^{\text{IP}}_{\text{GCV}}(t)$ is the concentration of GCV in the peritoneal cavity and $C^{\text{IP}}_0$ is the initial concentration of GCV in the peritoneal cavity. A summary of the system of ordinary differential equations consistent with the dose regimen of the gene therapy experiment is shown below:

$$\frac{dV}{dt} = G - N - S, V(0) = V_0,$$

$$G = \sigma \cdot V \cdot \left(1 - \frac{V}{K}\right), N = \lambda \cdot C^{P}_{GCV}, S = \frac{D_{Pep} \cdot C^{P}_{Pep}}{C^{P}_{Pep} + K_d([D_3])},$$

$$\frac{dC^{P}_{GCV}}{dt} = k^{GCV}_{a} \cdot C^{IP}_{GCV} - k^{GCV}_{ex} \cdot C^{P}_{GCV}, C^{P}_{GCV}(0) = 0,$$

$$\frac{dC_{GCV}^{IP}}{dt} = -k_a^{GCV} \cdot C_{GCV}^{IP}, C_{GCV}^{IP}(0) = 0,$$

$$C_{GCV}^{IP}(t) = \lim_{\varepsilon \to 0} C_{GCV}^{IP}(t - \varepsilon) + I_0, t = 12, 13, \ldots, 21$$

$$\frac{dC_{Pep}^{P}}{dt} = -k_{ex}^{Pep} C_{Pep}^{P}, C_{Pep}^{P}(0) = 0,$$

$$C_{Pep}^{P}(t) = \lim_{\varepsilon \to 0} C_{Pep}^{P}(t - \varepsilon) + I_1, t = 5.$$

In order to perform model parameterization, we began by sequentially fitting the model to the gene therapy data (*Figure 3C* and *Figure 4—figure supplement 1*) to estimate the unknown model parameters, which were later used to evaluate the pharmacodynamics of the soluble peptide given in treatment experiments (*Figure 3A* and *Figure 4—figure supplement 1*). To estimate the tumor growth rate constant $\sigma$, we solved *Equation 1* for $t$ in the range of 15– 21 days ignoring the terms $N$ and $S$, and used the data corresponding to GCV only (control) to drive a least squares optimization routine. Subsequently, we extrapolated tumor volume to time $t = 0$ to obtain $V_0$, and then used this initial condition to solve *Equation 1* for $t$ in the range of 0 to 21 days. At this step, the term $N$ was retained but $S$ was again set to zero, and the data corresponding to fd-AAVP-HSV*tk* + GCV group was then fit to estimate $\lambda$. Finally, the whole system including the term $S$ was solved for 0–21 days and the data corresponding to CSSTRESAC-AAVP-HSV*tk* + GCV group was used to estimate the parameters $D_{Pep}$, $K_d$, and $k_{ex}^{Pep}$. Further, the system of equations 6 and 7 was fit to the literature-derived plasma concentration kinetics of GCV after ip administration in mice to extract the unknown parameters $k_a^{GCV}$ and $k_{ex}^{GCV}$ for use in the tumor growth model. The computed parameters are shown in *Supplementary file 2*.

In the above calculations, we assumed a mouse weight of 20 g, and a volume of 10 ml per kg for the peritoneal cavity. The administered ip dose of GCV was 80 mg/kg/day, that is, $I_0 = 0.008$ mg/mm$^3$ and the iv administration of the peptide was $I_1 = 0.8$ mM. For all experiments, we used a carrying capacity $K$ of $10^4$ mm$^3$. To comply with animal testing regulations, the tumor-bearing mice were killed much before the value $10^4$ mm$^3$ was achieved; we however note that the allowable limit of tumor volumes during in vivo studies does not necessarily reflect the carrying capacity of the host. Hence, a literature-based value for $K$ was used (*Wu et al., 2018*). Of note, in the gene transfer experiments, the peptide is displayed on the AAVP particle, hence it has a different PK behavior (defined by $k_{ex}^{Pep}$) than the soluble CSSTRESAC peptide. Therefore, the excretion rate constant of the soluble CSSTRESAC peptide was refit while modeling its pharmacodynamics and was denoted as $k_{ex}^{Pep,sol}$.

In terms of predictions from the model, we show the fit for in vivo experiments performed in experimental mouse models plus a simulated clinical trial for treatment of human breast cancer with soluble CSSTRESAC (*Figure 4A* and *Figure 4—figure supplement 1*). The obtained value of $k_{ex}^{Pep,sol} = 38.0\,\text{day}^{-1}$, and the corresponding half-life of CSSTRESAC were computed as $t_{1/2}^{Pep,sol} = \ln 2/k_{ex}^{Pep,sol}$, which is ~26 min. Therefore, the half-life of the CSSTRESAC conforms to those of other targeting pharmacological data (*Pasqualini et al., 2015*) and indicates that soluble CSSTRE-SAC is rapidly cleared through renal excretion. To evaluate the importance of a possible competitive binding between 1,25-(OH)$_2$D$_3$ and soluble CSSTRESAC in the serum, the dissociation constant $K_d$ was perturbed by ± 20% of the reference parameter value. As such, an increase in $K_d$ would reflect the competitive binding of the antagonist 1,25-(OH)$_2$D$_3$, where the dissociation of CSSTRESAC from PDIA3 on the cell surface increases and the antitumor effects of CSSTRESAC decreases. Similarly, a reduced $K_d$ would reflect a stronger binding between CCSTRESAC and PDIA3 with the consequent inhibition of tumor growth (*Figure 4A*, *Figure 4—figure supplement 1*).

We also considered a hypothetical experiment in human patients where a constant rate of iv infusion of the soluble CSSTRESAC peptide was compared to the efficacy of a unit i.v. bolus. In this scenario, the kinetics of the soluble CSSTRESAC peptide is dictated by equation 8. The infusion constant $\mu$ denotes the asymptotic concentration of the peptide and $I(t)$ is the infusion rate. Our

mathematical model predicted that infusion of CCSTRESAC ($\mu = 0.75$) would resulted in greater reduction in tumor volume compared to bolus (*Figure 4A* and *Figure 4—figure supplement 1*). Thus, this in silico experiment clearly illustrates the use of a mathematical model in predicting potential therapeutic effects and limitations of soluble CSSTRESAC. Similarly, our mathematical model when fit to the experimental data showed satisfactory agreement as indicated by the Pearson correlation coefficient $R = 0.998$ (p = 0.001) (*Figure 4—figure supplement 1*). The model parameter estimates are listed (*Supplementary file 2*).

## PDIA3 gene expression in human single cells from TNBC patients

In order to evaluate the expression levels of *PDIA3* mRNA in human breast cancer samples, we obtained the clinical and scRNA-seq data from a publicly available single cell database from TNBC patients (BC07 - BC11) as originally reported (*Chung et al., 2017*). The scRNA-seq datasets were reported as TPM and were assessed through the GEO repository (accession number GSE75688). Gene expression levels of 35 pre-defined myeloid cells were retrieved and distributed into three groups according to *PDIA3* expression levels (set as high, medium, or low). A heat-map was generated to show potential associations between *PDIA3* and gene pathways characteristic of macrophages.

## Statistical analysis

Comparisons among the groups were assessed by One-way ANOVA with SigmaStat (SPSS Inc) and GraphPad Prism (GraphPad Software Inc). Statistical significance was set at a p-value of <0.05 unless otherwise specified. Normally distributed data are shown as bar graphs with means ± standard deviation (SD) or standard error of the mean (SEM) as indicated, whereas not normally distributed data are shown in box-and-whiskers plots: the boxes define the 25th and 75th percentiles, a line denotes the median and error bars define the 10th and 90th percentiles.

## Acknowledgements

This work was supported by the US DOD IMPACT grant W81XWH-09-1-0224 and by serial awards from the Gillson-Longenbaugh Foundation and the Susan G Komen Breast Cancer Foundation (to WA and RP). The RCSB Protein Data Bank is supported by grants to SKB from the National Science Foundation (DBI-1832184), the NIH (R01GM133198), and the US DOE (DE-SC0019749). This work has also been supported by the National Science Foundation Grant DMS-1930583 (ZW, VC), NIH Grants 1U01CA196403 (ZW, VC), 1U01CA213759 (ZW, VC), 1R01CA226537 (ZW, VC, WA, RP), 1R01CA222007 (ZW, VC), and U54CA210181 (ZW, VC). MCo was partially funded by FAPESP (2012/24105-3, 2020/13562–0). We thank Dr. Helen Pickersgill (Life Science Editors) for professional editing and Dr. Webster K Cavenee, Dr. Sylvia Christakos, and Dr. E Helene Sage for critical reading of the manuscript. JGG, WA, and RP are founders and equity stockholders of PhageNova Bio, which has licensed reagents disclosed in this manuscript. AH, WHPD, BP, WA and RP are entitled to royalty payments from this licensing agreement. RP is the Chief Scientific Officer and a paid consultant for PhageNova Bio. WA and RP are partially supported by a Sponsored Research Agreement from PhageNova Bio. AH, JGG, WA, and RP are inventors on issued and pending patent applications related to technology disclosed in this manuscript and will be entitled to royalties if licensing or commercialization occurs. WA and RP are also founders and equity holders of MBrace Therapeutics. RP serves as a Board Member of MBrace Therapeutics. These arrangements are managed and monitored in accordance with the established institutional conflict of interest policies of Rutgers, the State University of New Jersey. MCr is a consultant for CytoDyn, Sermonix Pharmaceuticals, G1 Therapeutics, Foundation Medicine, Dompé, ArcherDX and Menarini. MCr also receives honoraria/travel grants from Pfizer, Lilly, Novartis, Sermonix Pharmaceuticals, Foundation Medicine and Menarini. Other authors declare that they have no competing interests.

# Additional information

## Competing interests

Juri G Gelovani: is a founder and equity stockholder of PhageNova Bio, which has licensed reagents disclosed in this manuscript. JGG is also an inventor on issued and pending patent applications related to technology disclosed in this manuscript and will be entitled to royalties if licensing or commercialization occurs. Wadih Arap: is a founder and equity stockholder of PhageNova Bio, which has licensed reagents disclosed in this manuscript. WA is entitled to royalty payments from this licensing agreement. WA is partially supported by a Sponsored Research Agreement from PhageNova Bio. WA is an inventor on issued and pending patent applications related to technology disclosed in this manuscript and will be entitled to royalties if licensing or commercialization occurs. WA is also a founder and equity holder of MBrace Therapeutics. Renata Pasqualini: is a founder and equity stockholder of PhageNova Bio, which has licensed reagents disclosed in this manuscript. RP is entitled to royalty payments from this licensing agreement. RP is the Chief Scientific Officer and a paid consultant for PhageNova Bio. RP is partially supported by a Sponsored Research Agreement from PhageNova Bio. RP is an inventor on issued and pending patent applications related to technology disclosed in this manuscript and will be entitled to royalties if licensing or commercialization occurs. RP is also a founder and equity holder of MBrace Therapeutics. RP serves as a Board Member of MBrace Therapeutics. The other authors declare that no competing interests exist.

## Funding

| Funder | Grant reference number | Author |
|---|---|---|
| U.S. Department of Defense | W81XWH-09-1-0224 | Fernanda I Staquicini<br>Amin Hajitou<br>Wouter HP Driessen<br>Bettina Proneth<br>Marina Cardó-Vila<br>Daniela I Staquicini<br>Wadih Arap<br>Renata Pasqualini |
| Gilson Logenbaugh Foundation | | Fernanda Staquicini<br>Amin Hajitou<br>Wouter HP Driessen<br>Bettina Proneth<br>Marina Cardó-Vila<br>Daniela Staquicini<br>Wadih Arap<br>Renata Pasqualini |
| Susan G. Komen | | Fernanda Staquicini<br>Amin Hajitou<br>Wouter HP Driessen<br>Bettina Proneth<br>Marina Cardó-Vila<br>Daniela Staquicini<br>Wadih Arap<br>Renata Pasqualini |
| National Science Foundation | DBI-1832184 | Christopher Markosian<br>Zhihui Wang<br>Vittorio Cristini<br>Stephen Burley |
| National Institutes of Health | R01GM133198 | Fernanda Staquicini<br>Amin Hajitou<br>Wouter HP Driessen<br>Bettina Proneth<br>Marina Cardó-Vila<br>Daniela Staquicini<br>Zhihui Wang<br>Vittorio Cristini<br>Stephen Burley<br>Wadih Arap<br>Renata Pasqualini |

| U.S. Department of Energy | DE-SC0019749 | Stephen Burley |
| Fundação de Amparo à Pesquisa do Estado de São Paulo | 2012/24105-3 | Mauro Cortez |
| National Science Foundation | DMS-1930583 | Christopher Markosian<br>Zhihui Wang<br>Vittorio Cristini |
| National Institutes of Health | 1U01CA196403 | Fernanda Staquicini<br>Amin Hajitou<br>Wouter HP Driessen<br>Bettina Proneth<br>Marina Cardó-Vila<br>Daniela Staquicini<br>Zhihui Wang<br>Vittorio Cristini<br>Stephen Burley<br>Wadih Arap<br>Renata Pasqualini |
| National Institutes of Health | 1U01CA213759 | Fernanda Staquicini<br>Amin Hajitou<br>Wouter HP Driessen<br>Bettina Proneth<br>Marina Cardó-Vila<br>Daniela Staquicini<br>Zhihui Wang<br>Vittorio Cristini<br>Stephen Burley<br>Wadih Arap<br>Renata Pasqualini |
| National Institutes of Health | 1R01CA226537 | Fernanda Staquicini<br>Amin Hajitou<br>Wouter HP Driessen<br>Bettina Proneth<br>Marina Cardó-Vila<br>Daniela Staquicini<br>Zhihui Wang<br>Vittorio Cristini<br>Stephen Burley<br>Wadih Arap<br>Renata Pasqualini |
| National Institutes of Health | 1R01CA222007 | Fernanda Staquicini<br>Amin Hajitou<br>Wouter HP Driessen<br>Bettina Proneth<br>Marina Cardó-Vila<br>Daniela Staquicini<br>Zhihui Wang<br>Vittorio Cristini<br>Stephen Burley<br>Wadih Arap<br>Renata Pasqualini |
| National Institutes of Health | U54CA210181 | Fernanda Staquicini<br>Amin Hajitou<br>Wouter HP Driessen<br>Bettina Proneth<br>Marina Cardó-Vila<br>Daniela Staquicini<br>Zhihui Wang<br>Vittorio Cristini<br>Stephen Burley<br>Wadih Arap<br>Renata Pasqualini |
| Fundação de Amparo à Pesquisa do Estado de São Paulo | 2020/13562-0 | Mauro Cortez |

The funders had no role in study design, data collection and interpretation, or the decision to submit the work for publication.

## Author contributions

Fernanda I Staquicini, Conceptualization, Resources, Data curation, Software, Formal analysis, Supervision, Funding acquisition, Validation, Investigation, Visualization, Methodology, Writing - original draft, Project administration, Writing - review and editing; Amin Hajitou, Wouter HP Driessen, Christopher Markosian, Bedrich Eckhardt, Martin Trepel, Conceptualization, Data curation, Formal analysis, Investigation, Methodology, Writing - review and editing; Bettina Proneth, Marina Cardó-Vila, Daniela I Staquicini, Maria Hoh, Conceptualization, Data curation, Formal analysis, Investigation, Methodology; Mauro Cortez, Israel T Silva, Robin Anderson, Conceptualization, Formal analysis, Writing - review and editing; Anupama Hooda-Nehra, Mohammed Jaloudi, Richard L Sidman, Juri G Gelovani, Massimo Cristofanilli, Gabriel N Hortobagyi, Zaver M Bhujwalla, Stephen K Burley, Conceptualization, Writing - review and editing; Jaqueline Buttura, Diana N Nunes, Emmanuel Dias-Neto, Javier Ruiz-Ramírez, Prashant Dogra, Zhihui Wang, Conceptualization, Data curation, Formal analysis, Methodology, Writing - review and editing; Vittorio Cristini, Conceptualization, Formal analysis, Methodology, Writing - review and editing; Wadih Arap, Conceptualization, Resources, Formal analysis, Supervision, Funding acquisition, Validation, Visualization, Methodology, Writing - original draft, Project administration, Writing - review and editing; Renata Pasqualini, Conceptualization, Resources, Formal analysis, Supervision, Funding acquisition, Methodology, Writing - original draft, Project administration, Writing - review and editing

## Author ORCIDs

Fernanda I Staquicini (iD) https://orcid.org/0000-0003-1137-6575
Mauro Cortez (iD) http://orcid.org/0000-0001-6536-4647
Wadih Arap (iD) https://orcid.org/0000-0002-8686-4584

## Ethics

Animal experimentation: Eight-week-old female nude (nu/nu) mice and immunocompetent BALB/c mice were housed in the animal facilities of the University of Texas M.D. Anderson Cancer Center # 11-99-09935 and Rutgers University New Jersey Medical School (PROTO201800055), all in the USA. All animal procedures were reviewed and approved by the corresponding Institutional Animal Care and Use Committee (IACUC)-equivalent at each institution.

## Decision letter and Author response

Decision letter https://doi.org/10.7554/eLife.65145.sa1
Author response https://doi.org/10.7554/eLife.65145.sa2

# Additional files

## Supplementary files

- Supplementary file 1. MS/MS analysis of CSSTRESAC-binding proteins.
- Supplementary file 2. Computer parameters of the mathematical model.
- Transparent reporting form

## Data availability

All data generated or analysed during this study are included in the manuscript and supporting files. Source data files have been provided in Materials and methods and Supplementary files.

The following previously published dataset was used:

| Author(s) | Year | Dataset title | Dataset URL | Database and Identifier |
|---|---|---|---|---|
| Chung W, Eum HH, Lee HO, Lee KM, Lee HB, Kim KT, Ryu HS, Kim S, Lee JE, Park YH, Kan Z, Han W, Park WY | 2016 | Single-cell RNA-seq enables comprehensive tumour and immune cell profiling in primary breast cancer. | https://www.ncbi.nlm.nih.gov/geo/query/acc.cgi?acc=GSE75688 | NCBI Gene Expression Omnibus, GSE75688 |

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
