## [Decision Letter]

**Acceptance summary:**

This paper by Arap et al. utilizes phage display technology to identify a novel peptide that interacts with components of the Vitamin D receptor on tumor associated macrophages. To target this discovery to therapeutic advantage, a soluble peptide conjugated to cell suicide agents was created and found to suppress the growth of triple negative breast cancer cells in animal models.

**Decision letter after peer review:**

Thank you for submitting your article "Targeting a cell surface vitamin D receptor on tumor-associated macrophages in triple-negative breast cancer" for consideration by *eLife*. Your article has been reviewed by 2 peer reviewers, and the evaluation has been overseen by a Reviewing Editor and Mone Zaidi as the Senior Editor. The reviewers have opted to remain anonymous. The Reviewing Editor has drafted this to help you prepare a revised submission.

Summary:

Both reviewers noted the potential importance of the submission. However, both of them noted that some areas needed improvements, and specifically the role of vitamin D was highlighted. Given the extensive experimentation that both reviewers felt was largely sufficient for publication, a resubmission is encouraged focusing on addressing the relatively minor shortcomings noted below.

Essential Revisions:

Overall there are not a lot of significant revisions needed. The two listed below should be addressed for resubmission:

1. Both reviewers wanted additional information if the effects of peptide treatment are abrogated by excess vitamin D. Please ensure that evidence is included to address this topic.

2. Reviewer 1 aptly noted that the Discussion section is inadequate given the extensive data. Please expand the discussion to address questions brought up by both reviewers below. One additional topic for discussion not brought up by the reviewers, but that would be interesting is the potential to use this therapy in combination with other immunomodulatory drugs, which now have broad multi-tumor applications(i.e. those targeting the PD-L1 axis). Would is be possible to combine the two therapies to be synergistic?

*Reviewer #1:*

Overall Strengths: The paper has many strengths. Overall, it presents a novel function for a specific vitamin D receptor on type II tumor associated macrophages (Type II TAMS), a finding not previously made. It further provides experimental evidence that supports the possiblilty of this discovery having an eventual impact on the treatment of patients with Triple Negative Breast Cancer.

Importantly, the authors provide evidence that a specific nonapeptide CSSTRESAC activates the PDIA3 vitamin D receptor on TAMs, and further stimulates a pro-inflammatory cytokine response, likely further stimulating tumor aggressiveness.

Based on this novel discovery, the authors propose multiple approaches to interfering with the pro-tumorigenic Vitamin D-receptor TAM axis in Triple Negative Breast Cancer, an aggressive form of this disease. These include 1. Using a soluble form of the nonapeptide to blunt tumor growth, and 2. Using the peptide to target cell suicide moieties to the tumor. Both of these approaches proved effective at retarding tumor growth in the data presented in the manuscript and both represent novel approaches to the clinical management of Triple Negative Breast Cancer if they can be shown to be safe and effective. The targeting approach is of particular interest due to the continued success of the prostate-membrane-specific antigen (PSMA) in targeting both imaging radionuclides and antibody-drug conjugates to tumor sites. This type of tumor targeting is clearly gaining traction in clinical cancer therapy and new targeting modalities are sorely needed. The presentation of CSSTRESAC as a novel targeting agent for Triple Negative Breast Cancer and, one would expect, for other tumors with a strong M2 TAM response is potentially quite significant.

Technical Strengths:

This team leaders have worked on the use of phage display technology to identify biologically active peptides for many years and are certainly leaders in this field. The identification of CSSTRESAC as a vitamin D receptor ligand is a significant scientific advancement that would have been difficult to accomplish by other means.

The tumor experiments are well done and provide compelling evidence that use of the peptide can retard tumor growth in preclinical experiments.

The targeting experiments, both with phage and with conjugated peptide, provide confidence that this peptide, or future derivatives could be utilized to direct reagents to TAM infiltrated tumors, a potentially important clinical use.

Potential Weaknesses:

The possibility of toxicity during prolonged treatment with the CSSTRESAC peptide or its conjugates could be investigated more thoroughly,

Preliminary experiments suggest that binding of the receptor to the receptors is not stronger than the natural vitamin D components. Will this compromise the potential clinical utility of the approach?

Comments for authors:

The authors provide evidence that liver enzymes are not adversely affected by treatment with the peptide. They should also show weights of the animals treated with the peptide during the course of the tumor experiments.

The authors provide convincing evidence that PDIA3 expression on monocytes is a mediator of CSSTRESAC activity, and that peptide-expressing phage bind to PDIA3 and DBP. It would be worthwhile if they could show directly the affinity of the peptide binding to purified DBP and PDIA3, perhaps by nuclear plasmon resonance or another direct biochemical analysis.

In Figure 3, viral gene therapy appears to give a greater anti -TAM response but not a greater anti-tumor response. Is this a meaningful difference?

The authors provide computational molecular modeling to determine whether the nonapeptide is conformationally similar to vitamin D and determine that it should bind directly to DBP. As this reviewer is not sufficiently versed in that technology, others must weigh in on this data.

Inadequate Discussion:

While it is common to see manuscript discussions where the overlong conclusions are not justified by the amount or importance of the results, this paper represents the opposite case. Surely a manuscript with 8 plus pages of results, 14 pages of methods and approximately 60 data panels deserves more than one paragraph of discussion. There are many aspects of this paper where it would be worthwhile to hear the author's interpretation. Just a few of these are listed here, but I'm sure that other readers would want more.

a. Why a cyclic peptide vs a linear peptide?

b. What is the relevance to Vit D activity in macrophages in other contexts including inflammation and atherogenesis?

c. PDIA3 null mice were reported to show severe bone abnormalities. If true, would this impact the potential use of inhibiting this axis in patients with metastatic breast cancer where bone integrity might be compromised?

d. If the effects of peptide treatment are abrogated by excess vitamin D, would it be necessary to restrict vitamin D intake in patients undergoing treatment with soluble peptide?

e. Will determination of TAM infiltration be necessary to preselect patients who might be candidates for peptide therapy?

*Reviewer #2:*

This manuscript focuses on using phage display-based approach to identify peptides that target tumor associated macrophages to treat triple negative breast cancers (TNBC) and potentially other tumors that express the cell surface receptor PDIA3. The authors did an extensive and well-organized study that identified a peptide (CSSTRESAC) that appears as a mimetic to active vitamin D, and was found to bind to the vitamin D receptors DBP and PDIA3, the later of which is expressed on the surface of tumor macrophages (TAMs). They show that CSSRTESAC enhances a pro-inflammatory response that is tumor-inhibiting and thus could become a lead drug target.

The manuscript provides extensive supplementary data to well-support their investigations, and thus there are few short-comings. The one major area of weakness stems from the hypothesis that the CSSRTESAC peptide is serving as a vitamin D mimetic to exert its tumor-inhibiting effects. In Figure 2 the authors demonstrate that CSSRTESAC can induce a marked pro-inflammatory phenotype, however such effects are not seen by vitamin D, which the authors find could actually competitively inhibit the pro-inflammatory response. This interplay between CSSRTESAC and vitamin D was not explored in vivo unfortunately.

This is an interesting topic that identifies a potential immunomodulatory target that could be targeted. This has the potential to have a profound impact on oncology and thus overall with some revision should deserve publication.

Overall this is an outstanding and well-supported manuscript.

1. The one major area of weakness stems from the hypothesis that the CSSRTESAC peptide is serving as a vitamin D mimetic to exert its tumor-inhibiting effects. In Figure 2 the authors demonstrate that CSSRTESAC can induce a marked pro-inflammatory phenotype, however such effects are not seen by vitamin D, which the authors find could actually competitively inhibit the pro-inflammatory response. This interplay between CSSRTESAC and vitamin D was not explored in vivo. More mechanistic studies would have significantly strengthened the paper. Why does CSSRTESAC binding to PDIA3 induce a profound pro-inflammatory, tumor inhibiting response but stronger binding by vitamin D does not?

2. The main hypothesis is that CSSRTESAC induces changes in macrophages that lead to a pro-inflammatory response inhibiting tumor growth. What happens when the inflammatory response is inhibited? For example steroids are often used in many chemo regimens; what is the effect of CSSRTESAC on tumor growth when combined with steroids vs steroids alone? This again goes back to the question on mechanism.

3. There is a typo on line 210 of page 10: "macrophage" should read "macrophages"

4. Not sure if line 369 page 17 has a typo: "incubated ON at". Perhaps this is correct, as I am not familiar with the methodology.

---

## [Author Response]

Essential Revisions:Overall there are not a lot of significant revisions needed. The two listed below should be addressed for resubmission:1. Both reviewers wanted additional information if the effects of peptide treatment are abrogated by excess vitamin D. Please ensure that evidence is included to address this topic.

Thank you for the opportunity to address this ligand-receptor interaction. We have now expanded our original Results and Discussion sections to emphasize the mathematical modeling projections used to assess the competitive nature of the binding interactions observed among the new cyclic ligand peptide CSSTRESAC, 1,25-(OH)_2_D_3_, and vitamin D-binding protein (VDB), along with their potential biochemical consequences for targeting in vivo. Our model accounts for two primary opposing processes: tumor cell growth and death caused by the CSSTRESAC peptide, while also allowing for competitive antagonism exhibited by 1,25-(OH)_2_D_3_ in serum. If one assumes an adult mouse weight of ~20 g, and a volume of ~10 mL/kg for the peritoneal cavity, normal levels of total vitamin D in the peripheral blood are estimated at ~50 ng/mL, which is ~150 nM, of which only ~1% would be free to interact with CSSTRESAC. As such, at 10 mg/kg treatment dose, serum peak concentrations of soluble CSSTRESAC peptide are estimated at the μM range, and therefore unlikely to be competed off by the circulating 1,25-(OH)_2_D_3_. Moreover, we have now used a mathematical model to simulate a clinical trial of human breast cancer patients with soluble CSSTRESAC and the importance of a potential competitive binding between 1,25-(OH)_2_D_3_ and CSSTRESAC in the serum (Figure 5 and Supplementary Figure 8). By comparing intravenous (IV) administration methods (i.e., infusion vs. bolus), our model predicted that an infusion setting would result in ~400 mm^3^ greater reduction in tumor volume compared to bolus administration, without interference from the competing native 1,25-(OH)_2_D_3_ in serum. These data have now been added to the revised Results (pp. 12-13) and revised Discussion (pp. 14-16).

2. Reviewer 1 aptly noted that the Discussion section is inadequate given the extensive data. Please expand the discussion to address questions brought up by both reviewers below.

We appreciate this insightful suggestion. We have now expanded the Discussion section of the original manuscript to incorporate the points raised by Reviewer 1 and have also discussed the potential use of CSSTRESAC-targeted therapies in combination with immunomodulatory drugs. These revisions can now be found in the revised manuscript (pp. 14-16).

Reviewer #1:[…] Potential Weaknesses:The possibility of toxicity during prolonged treatment with the CSSTRESAC peptide or its conjugates could be investigated more thoroughly,

Thank you. This has now been added to the revised Discussion (p. 14-16).

Preliminary experiments suggest that binding of the receptor to the receptors is not stronger than the natural vitamin D components. Will this compromise the potential clinical utility of the approach?

This is an important point. We have now introduced an in silico model to address the potential effects of serum 1,25-(OH)_2_D_3_ on the therapeutic activity of soluble CSSTRESAC when administered into the circulation (revised Results and Discussion). Please see our more detailed response to point 1 of the Reviewing Editor.

Reviewer #2:[…] 1. The one major area of weakness stems from the hypothesis that the CSSRTESAC peptide is serving as a vitamin D mimetic to exert its tumor-inhibiting effects. In Figure 2 the authors demonstrate that CSSRTESAC can induce a marked pro-inflammatory phenotype, however such effects are not seen by vitamin D, which the authors find could actually competitively inhibit the pro-inflammatory response. This interplay between CSSRTESAC and vitamin D was not explored in vivo. More mechanistic studies would have significantly strengthened the paper. Why does CSSRTESAC binding to PDIA3 induce a profound pro-inflammatory, tumor inhibiting response but stronger binding by vitamin D does not?

This is indeed an intriguing observation. Our current interpretation is that a pharmacological dose of soluble CSSTRESAC administered IV during treatment is likely to result in differential effects in PDIA3-expressing TAMs, which would not be observed when PDIA3 is exposed to physiological amounts of circulating 1,25-(OH)_2_D_3_. Alternatively, one might speculate that 1,25-(OH)_2_D_3_ could perhaps lead to stimulation of both membrane- and nuclear-initiated steroid signaling while CSSTRESAC only stimulates membrane-initiated signaling pathways. Future studies will further clarify between these possible biochemical scenarios.

2. The main hypothesis is that CSSRTESAC induces changes in macrophages that lead to a pro-inflammatory response inhibiting tumor growth. What happens when the inflammatory response is inhibited? For example steroids are often used in many chemo regimens; what is the effect of CSSRTESAC on tumor growth when combined with steroids vs steroids alone? This again goes back to the question on mechanism.

This is a complex and ongoing line of mechanistic investigation that is now discussed in the revised manuscript. Future experiments will certainly shed light on the functional role(s) of the inflammatory response, but we think these are beyond the scope of the original discovery and early translational work reported here.